# The Aurora B specificity switch is required to protect from non-disjunction at the metaphase/ anaphase transition

Joanna R. Kelly [1,6], Silvia Martini[1], Nicola Brownlow[1,7], Dhira Joshi[2], Stefania Federico[2], Shirin Jamshidi [3], Svend Kjaer [4], Nicola Lockwood[1], Khondaker Miraz Rahmen [3], Franca Fraternali[5], Peter J. Parker[1,3✉] & Tanya N. Soliman [1,8✉]

The Aurora B abscission checkpoint delays cytokinesis until resolution of DNA trapped in the cleavage furrow. This process involves PKCε phosphorylation of Aurora B S227. Assessing if this PKCε-Aurora B module provides a more widely exploited genome-protective control for the cell cycle, we show Aurora B phosphorylation at S227 by PKCε also occurs during mitosis. Expression of Aurora B S227A phenocopies inhibition of PKCε in by-passing the delay and resolution at anaphase entry that is associated with non-disjunction and catenation of sister chromatids. Implementation of this anaphase delay is reflected in PKCε activation following cell cycle dependent cleavage by caspase 7; knock-down of caspase 7 phenocopies PKCε loss, in a manner rescued by ectopically expressing/generating a free PKCε catalytic domain. Molecular dynamics indicates that Aurora B S227 phosphorylation induces conformational changes and this manifests in a profound switch in specificity towards S29 TopoIIα phosphorylation, a response necessary for catenation resolution during mitosis.

[1] Protein Phosphorylation Laboratory, Francis Crick Institute, 1 Midland Rd, London NW1 1AT, UK. [2] Peptide Chemistry Platform, Francis Crick Institute, 1 Midland Rd, London NW1 1AT, UK. [3] School of Cancer and Pharmaceutical Sciences, King's College London, London, UK. [4] Structural Biology Platform, Francis Crick Institute, 1 Midland Rd, London NW1 1AT, UK. [5] Randall Centre for Cell and Molecular Biophysics, King's College London, London, UK. [6]Present address: Cancer Research UK, Manchester Institute, Alderley Park SK10 4TG, UK. [7]Present address: Instituto de Neurociencias, Av. Santiago Ramón y Cajal s/n 03550, San Juan de Alicante, Spain. [8]Present address: Barts Cancer Institute, Queen Mary University London, Charterhouse Square, London EC1M 6BE, UK. ✉email: peter.parker@crick.ac.uk; t.soliman@qmul.ac.uk

Entry to and exit from mitosis is exquisitely controlled to ensure faithful propagation of the resultant daughter cells. Genome-protective pathways such as the MiDAS salvage pathway, engaged upon the detection of incomplete DNA replication at mitosis[1], or the non-disjunction response[2] have evolved in addition to cell cycle checkpoints to ensure the fidelity of division.

As a master regulator of mitosis, Aurora B has been ascribed functions throughout mitosis and cytokinesis[3]. We have previously described a switch in Aurora B substrate specificity after second site phosphorylation of the activation loop at S227[4]. Upon retention of chromatin in the cytokinetic furrow an Aurora B-dependent abscission checkpoint is engaged and protein kinase C epsilon (PKCε) is required for exit once resolution of the lagging DNA strands occurs. Specifically, PKCε phosphorylates Aurora B at S227 which induces a switch in Aurora B substrate specificity, such that it then phosphorylates the adaptor protein Borealin at S165[4] to release the ESCRTIII component CHMP4C from a closed, inactive conformation to allow polymerisation of the ESCRTIII complex and successful completion of abscission[5,6].

PKCε-dependent pathways provide alternative, failsafe mechanisms for passage through mitosis. A conditional requirement for PKCε has been reported for bipolar spindle assembly[7] and the metaphase response to catenation[2] in a subset of transformed cell lines which lack a competent topoisomerase IIα (TopoIIα)-dependent G2 arrest. During metaphase, if the cell is challenged by excess catenation, PKCε instigates a delay in anaphase entry through delayed spindle assembly checkpoint (SAC) silencing and mitigates dynein directed streaming of the RZZ complex[2]. The engagement of PKCε in these TopoIIα-dependent G2 arrest deficient cells occurs by an as yet, undefined mechanism.

Here we assess the penetrance of PKCε-Aurora B module action in responding to genomic stress by addressing the requirement at the metaphase–anaphase transition. We demonstrate that the switch in Aurora B substrate repertoire driven by PKCε-mediated phosphorylation of S227 during cytokinesis appears to also function during metaphase. This correlates with the Caspase-7 dependent proteolytic activation of PKCε. The phosphorylation of Aurora B at S227 induces a conformational change in the activation loop of the kinase and is required for the switch in substrate specificity of Aurora B. This allows selection of substrates required for satisfaction of the SAC and enhances TopoIIα decatenation through phosphorylation of S29 promoting sister chromatid disjunction prior to anaphase entry.

## Results

**Chromosome disjunction requires Aurora B S227 phosphorylation.** To assess the dependence on the phosphorylation of Aurora B S227 for mitotic fidelity we introduced the non-phosphorylatable GFP-Aurora B S227A mutant into DLD1 cells. This results in an increase in the number of cells with aberrantly shaped nuclei (Fig. 1a) suggesting chromosome non-disjunction events, consistent with previous reports of an increase in binucleate cells observed on expression of Aurora B S227A[4]. There is also an increase in DAPI-positive bridges and PICH-positive strands present in anaphase cells (Fig. 1b). This bridging phenotype is associated with unresolved sister chromatid intertwines, indicative of catenation[8]. Direct assessment of this by removal of centromeric cohesion through knockdown of Shugoshin 1 (Sgo1) (Fig. 1c), showed that a significant increase in retained sister chromatid pairs was observed (Fig. 1d), comparable with knockdown of TopoIIα. We have previously demonstrated that this association between sister chromatid pairs is dependent on TopoIIα as post-extraction treatment of metaphase spreads with recombinant TopoIIα results in rescue of this phenotype to control levels[2].

A PKCε-dependent pathway responds to the retention of metaphase catenation in transformed cells with a defective TopoIIα-dependent G2 checkpoint[2]. On TopoIIα inhibition with ICRF193, DLD1 cells delay anaphase entry and in the absence of PKCε[2] or with the expression of Aurora B S227A (Fig. 1e) this delay is compromised, consistent with the idea that Aurora B is operating on this pathway downstream of PKCε. In the presence of catenation, inhibition of PKCε activity in a mitotically enriched population of cells (Supplementary Fig. 1a) with either BLU577 (Fig. 1f) or BIM1 (Supplementary Fig. 1b) resulted in a significant reduction in Aurora B S227 phosphorylation by immunoblot. Notably, there is no change in phosphorylation of Aurora B S227 in asynchronously growing cultures treated with PKC inhibitors (Supplementary Fig. 1c), indicating that under these conditions, this site is phosphorylated by an as yet unidentified kinase. The reduction in phosphorylation in mitotic extracts that we observe with PKC inhibition shows that this PKCε-Aurora B regulatory module is cell cycle dependent, occurring in mitosis, as well as cytokinesis as previously described[4].

Expression of the non-phosphorylatable Aurora B mutant does not preclude the SAC response in these cells as they still mount a checkpoint arrest in the presence of the microtubule assembly inhibitor nocodazole (Fig. 1g). The metaphase catenation response is characterised by MAD2$^{low}$/Bub1$^{high}$/BubR1$^{high}$ kinetochore staining, which is sensitive to PKCε loss or inhibition[2]. After ICRF193 treatment of DLD1 cells, BubR1 is maintained at the kinetochore in the Aurora B wild type expressing cells but not in those expressing Aurora B S227A, accounting for the delay in anaphase entry and bypass of the delay respectively (Fig. 1h). Nevertheless, cells expressing Aurora B S227A mount a SAC response in response to nocodazole inducing rapid relocalisation of BubR1 to the kinetochore in these cells. Consistent with a normal implementation of the SAC, MAD2 and BubR1 are localised to the kinetochore during prometaphase in these cells, but MAD2 is absent in metaphase cells irrespective of ICRF193 treatment or expression of Aurora B S227A (Fig. 1i).

**A chromatin-associated pool of PKCε is cleaved in mitosis.** We sought to determine the mechanism by which the PKCε-Aurora B regulatory module operates at the metaphase–anaphase transition. We find there are sub-populations of GFP-PKCε present; a centrosomal, spindle-associated pool that we have previously described[7] as well as a chromatin-associated compartment that can be visualised by immunofluorescence (Fig. 2a) and appears to be enriched during metaphase. Upon fractionation of mitotic cell extracts, it was apparent that within this latter compartment GFP-PKCε was partially fragmented, as evident in the appearance of two faster migrating, immunoreactive bands at 35 and 43 kDa recognised by a C-terminal PKCε antibody (Fig. 2b). An N-terminal cleavage fragment could be detected also by probing for GFP (Fig. 2b). PKCε has been previously reported to be phosphorylated during mitosis at S346, S350, S368 to facilitate 14-3-3 binding[9], resulting in slower migration of full length PKCε bands by western blot of the TritonX-100 soluble fraction (Supplementary Fig. 2a). The lower molecular weight species observed in the insoluble fraction were verified as PKCε as these bands could no longer be detected by western blotting after RNAi-mediated depletion of PKCε (Supplementary Fig. 2a). We demonstrate mitosis-specific cleavage of both endogenous (Supplementary Fig. 2b) and exogenously expressed (Supplementary Fig. 2c) PKCε across multiple cell lines, indicating that this is a widespread event. Some cleavage of PKCε is evident in asynchronously cycling cells and we attribute this observation to the presence of mitotic cells in these cultures. The observed cleavage of PKCε is chromatin associated (Supplementary Fig. 2d) and

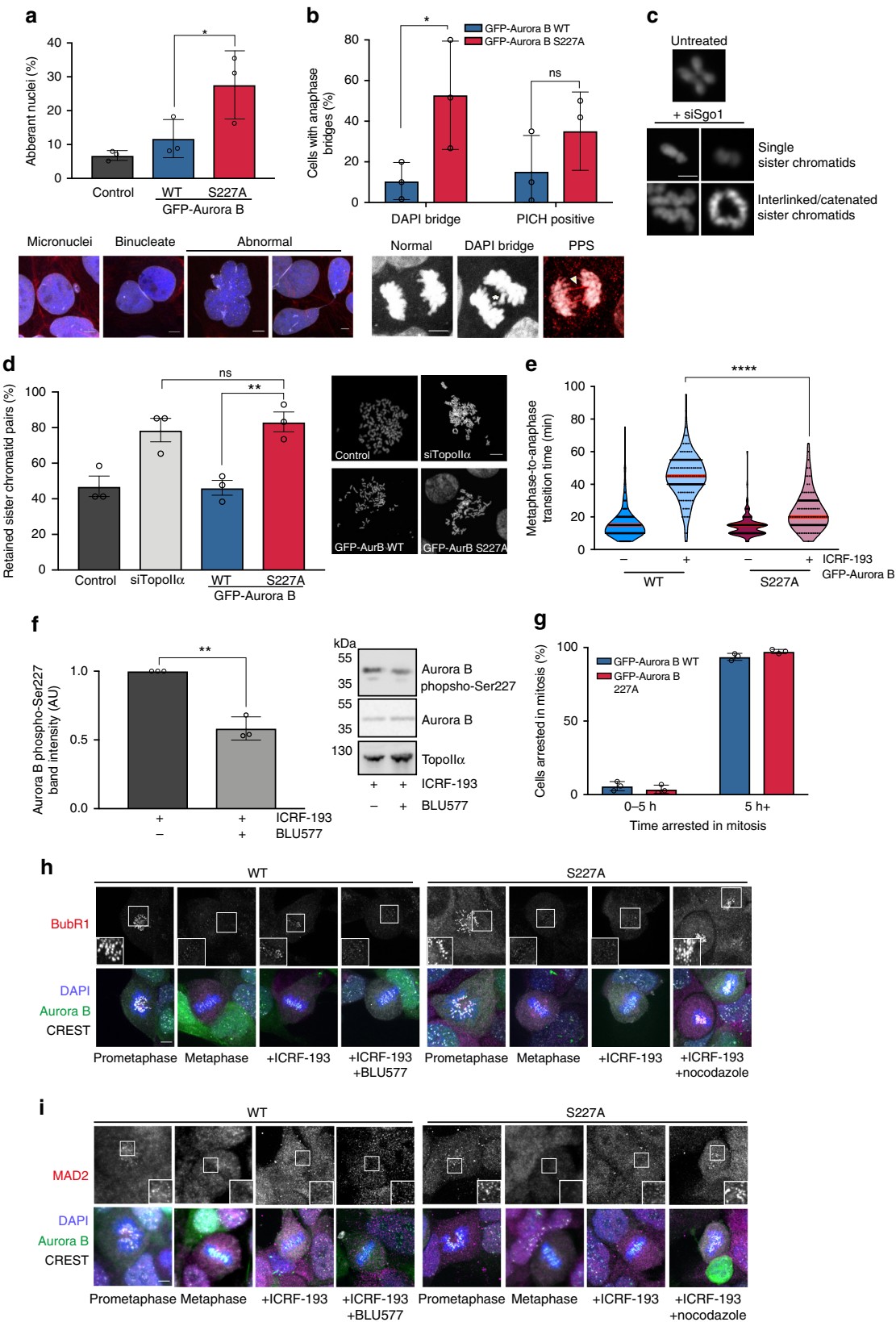

mitosis specific, evidenced by an enrichment of these cleavage products in prometaphase synchronised cells (Supplementary Fig. 2e). Enrichment in G1/S or G2 phase did not yield these faster migrating cleaved species (Supplementary Fig. 2f). Inhibition of PKCε using the selective inhibitor BLU577[2,4,10] did not influence proteolytic cleavage nor did inhibition of TopoIIα, known to engage a PKCε dependent mitotic pathway[2] (Supplementary Fig. 2g). The fragment produced is characterised by phosphorylation on key priming sites and these modifications are essential for catalytic competence (Supplementary Fig. 2h)[11].

**Fig. 1 Phosphorylation of Aurora B S227 is required to support the metaphase response to catenation. a** DLD1 cells expressing GFP-Aurora B wild type or S227A were scored for nuclear morphology. The mean ± SD of three independent experiments is graphed (n = ≥200) (one-way ANOVA, *P = ≤0.024; Control 6.76% ± 1.46, WT 11.76% ± 5.63, S227A 27.63% ±10.07). Scale bar = 5 μm. **b** Anaphase cells were scored for DAPI-positive bridges or PICH-positive strands. The graph represents the mean ± SD of three independent experiments, >20 cells per condition were scored per experiment (two-tailed Students' *t* test, *P = 0.027, DAPI: WT 10.35% ± 7.49, S227A 46.02% ± 25.62, PICH: WT 17.18% ± 14.8, S227A 35.98% ± 15.83). Scale bar = 5 μm. **c** Representative images of single sister chromatids or retained/catenated sister chromatids after 24 h Shugoshin knockdown (siSgo). Scale bar = 2 μm. **d** Cells were scored for the percent retained (catenated) sister chromatids. The mean ± SEM of three experiments where ≥15 cells per experiment were analysed is graphed (one-way ANOVA; **P = 0.0061; siControl 47.05% ± 5.75, siTopoIIα 78.84 ± 6.61, WT 46.23 ± 4.13, S227A 83.21 ± 5.561). Scale bar = 5 μm. **e** DLD1 cells expressing GFP-Aurora B wild type or S227A were treated ±1 μM ICRF193 immediately prior to time-lapse imaging and scored for time from metaphase to anaphase. Violin plots represent all cells scored, median in red. A minimum of 100 cells per condition in three independent experiments were analysed. (one-way ANOVA; ****P = ≥0.0001). **f** Immunoblot of DLD1 cells (representative of three independent experiments) enriched in mitosis and treated with ICRF193 (1 μM) ± BLU577 (500 nM). TritonX-100 insoluble fractions were probed for Aurora B phospho-S227, total Aurora B, and TopoIIα. Mean ± SD of the immunoblots is graphed (two-tailed Students' *t* test; **P = 0.001; +ICRF/+BLU 0.5835 ± 0.085). **g** Cells were treated with 500 nM nocodazole and imaged using live-cell time-lapse microscopy for 24 h and scored for their time arrested in mitosis. The mean ± SD of three experiments where >50 cells were analysed per condition, per experiment are graphed. Representative images of (**h**) BubR1 and (**i**) MAD2 kinetochore localisation after inducing a metaphase delay with ICRF193 (4 h, 1 μM) ± BLU577 (20 min 500 nM) or ±nocodazole (5 min, 500 nM), from three independent experiments, a minimum of ten images were acquired. DAPI (blue), GFP-Aurora B (green), BubR1/MAD2 (magenta), CREST (white). Scale bar = 5 μm.

**Non-canonical Caspase-7 cleavage of PKCε.** Previously, PKCε has been reported to be cleaved by Caspase-7 in vitro and this is associated with a pattern of fragmentation consistent with that observed here[12]. The sites of cleavage were mapped to D383, located in the V3 hinge region, and D451, in the α-C helix of the kinase domain (see schematic, Fig. 2c). The consequence of cleavage at the D383 site is the liberation of a free kinase domain which relieves PKCε from auto-inhibition. We therefore sought to examine whether this was the mechanism of cell cycle-dependent PKCε cleavage in vivo. Cleavage of PKCε to the 43 kDa free kinase domain was significantly inhibited by treatment of cells with the pan-Caspase inhibitor z-vad-FMK (Supplementary Fig. 3a). Knockdown of Caspase-7 resulted in a 64% reduction of the cleaved, TritonX-100 insoluble PKCε species, however no significant reduction in cleavage was observed with depletion of Caspase-3 (Fig. 2d), suggesting a predominantly Caspase-7 dependent pathway, in agreement with observations by Basu et al.[12]. We note here, that the cleavage is not an equilibrium event it is unidirectional, hence it is difficult to achieve a more penetrant blockade as the residual activity ratchets towards cleavage.

We observed no evidence of an apoptotic programme being triggered associated with or consequent to Caspase-7 cleavage of PKCε. Treatment of cells with ICRF193 or nocodazole did not affect cell viability (Supplementary Fig. 3b) nor did it induce cleavage of the apoptotic marker, poly (ADP ribose) polymerase (PARP) (Supplementary Fig. 3c). We observe cleavage of Caspase-7 to its active form in cells arrested in prometaphase but not those enriched in G1/S or G2 phases (Supplementary Fig. 3d).

We then investigated whether the Caspase-7-dependent cleavage of PKCε was required to sustain the metaphase catenation response. It was found that Caspase-7 knockdown led to a substantial loss of the metaphase delay (Fig. 2e), similar to that observed with expression of the Aurora B S227A mutant (Fig. 1e). Importantly, this was rescued by the expression of the kinase domain of PKCε indicating that this is the necessary pathway operating under the control of Caspase-7 (Fig. 2f). Collectively the data indicate an apoptosis-independent role for Caspase-7 of which PKCε is the critical target in this cell cycle pathway and that cleavage of PKCε to release the kinase domain is required for the ICRF193-induced delay in the metaphase–anaphase transition.

**PKCε cleavage protects from chromosome non-disjunction.** Mutation of the aspartate residues identified as Caspase-7

cleavage sites by Basu et al.[12] to asparagine, demonstrated that these are also the sites of mitotic cleavage (Fig. 3a). The expressed cleavage-site mutants were phosphorylated on the PKCε priming sites indicating that loss of catalytic competence did not appear to underlie the non-cleavage (Supplementary Fig. 4a, b). We then assessed the biological significance of this active catalytic fragment of PKCε that had been cleaved at D383 and in particular its potential role in the metaphase catenation-associated cell cycle delay, since we know that this pathway shows dependence on PKCε.

We engineered a DLD1 cell line to express inducible, GFP-tagged PKCε where the cleavage sites have been mutated (Supplementary Fig. 4c). Endogenous PKCε is depleted in experiments where GFP-PKCε is induced to ensure results are not confounded by the dominant behaviour of the cleaved fragment. These cell lines were characterised and expression of the GFP-PKCε mutants does not affect the SAC (Supplementary Fig. 4d) nor induce PARP cleavage (Supplementary Fig. 4e), further suggesting the observed cleavage of PKCε was independent of a pro-apoptotic role of Caspase-7. By blocking the cleavage at D383 (with or without a block to cleavage at D451), we observed an increase in the number of cells exhibiting aberrantly shaped nuclei (Fig. 3b). Loss of cleavage was also associated with an increase in both DAPI-positive chromosome bridges and PICH-positive strands, indicative of chromosome non-disjunction errors (Fig. 3c), phenocopying Aurora B S227A expression. Furthermore, cleavage of PKCε was required to support resolution of catenation (Supplementary Fig. 4f), as the non-cleavable PKCε failed to promote resolution of catenation in metaphase (Fig. 3d). We cannot conclusively exclude the presence of ultra-fine bridges or retention of cohesin as a contributing cause of sister chromatid non-disjunction due to technical limitations of available reagents; however, the presence of non-disjoined sister chromatids was rescued in vitro with recombinant TopoIIα, as we have demonstrated previously[2]. The increment of retained sister chromatid pairs that is reversed by treatment with recombinant TopoIIα in vitro provides a robust functional assay for the extent of association dependent upon catenation. This is certainly not 100% of all non-disjunction events which may indeed include other modes of sister chromatid association such as a cohesin-removal defect[13,14], but these are not impacted by the PKCε-Aurora B pathway reported here.

**The metaphase catenation response requires cleaved PKCε.** In cell lines where cleavage to the free kinase domain was prevented (D383N and D383/451N mutants), a loss of the characteristic

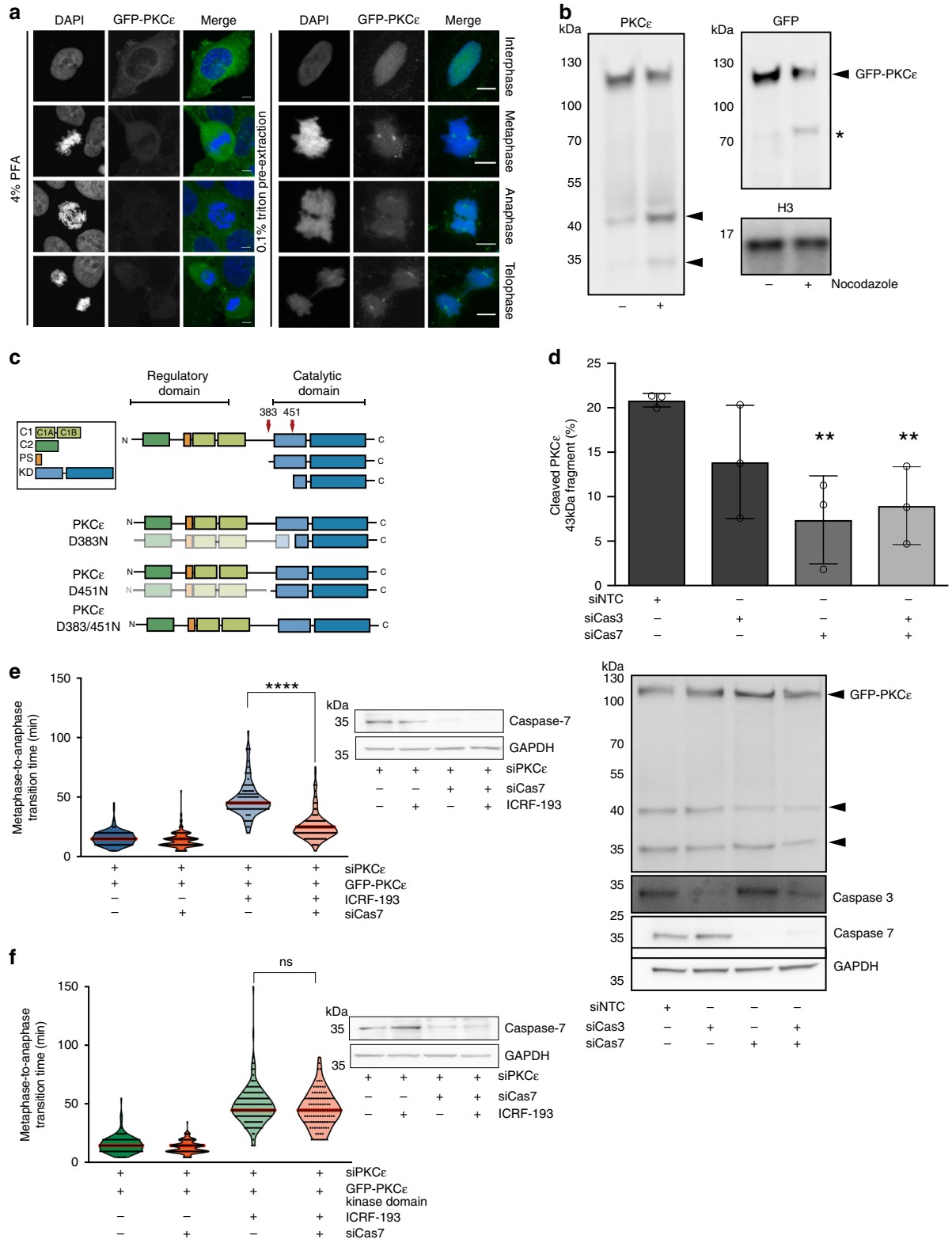

ICRF193-induced delay to anaphase entry was observed (Fig. 3e). To assess whether the cleavage product of PKCε is sufficient to support the metaphase functions of PKCε, we expressed the PKCε kinase domain in cells where the endogenous PKCε had been depleted (Fig. 3f, Supplementary Fig. 4g). Interestingly, the catalytic domain of PKCε is sufficient to maintain both the catenane

resolution pathway (Supplementary Fig. 4f) and the ICRF193-induced delay to anaphase entry (Fig. 3f). The cleavage product was able to fully recover PKCε functionality in both pathways after siRNA knockdown and there was no change to the meta-phase duration or the number of catenated sister chromatids compared with the control. This suggested an essential role for

**Fig. 2 A chromatin-associated pool of PKCε is cleaved by Caspase-7 in mitosis. a** Pre-extraction of cells with 0.1% Triton X-100 prior to fixation revealed a chromatin-associated pool of PKCε. Images are representative of three independent experiments. Scale bar = 5 μm. **b** Triton X-100 insoluble fraction of asynchronous or nocodazole (500 nM) arrested DLD1 cell lysates were probed for PKCε, GFP, and Histone H3. Western blot is representative of three independent experiments. **c** Schematic of PKCε domains and cleavage products with putative cleavages sites described by Basu et al.[12] marked by red arrows. **d** Immunoblot of GFP-PKCε to assess cleavage after knockdown of Caspases 3 and 7. Graph denotes the mean ± SD of three independent experiments where the proportion of cleaved 43 kDa fragment of PKCε present in the immunoblots was determined by densitometry (two-tailed Students' $t$ test; **$P = 0.01$; siControl 20.85 ± 0.76, SiCas3 13.9 ± 6.38, siCas7 7.4 ± 4.95, siCas3 + siCas7 8.98 ± 4.37). DLD1 cells were depleted of Caspase-7 and induced to express (**e**) GFP-PKCε wild type or (**f**) GFP-PKCε kinase domain and assessed for their response to metaphase catenation. Cells were treated with ±1μM ICRF193 immediately prior to time-lapse imaging and scored for the time to progress from metaphase to anaphase. Violin plots represent all cells scored, the median is in red. A minimum of 100 cells per condition in three independent experiments were analysed. (One-way ANOVA, ****$P = ≤0.0001$). Caspase-7 knockdown was confirmed by immunoblot for each experiment.

PKCε catalytic fragment production in execution of both the catenation-associated metaphase delay and also a catenation resolution pathway.

To address any potential confounding effect associated with the prolonged, constitutive expression of this kinase domain, we established expression of a rapamycin-inducible TEV-cleavage mutant of PKCε that permitted timed cleavage of the protein (Fig. 3g, h)[15,16]. As for the constitutively expressed kinase domain, metaphase induced cleavage was able to support the delay to anaphase entry (Fig. 3i). Hence the cell cycle associated cleavage of PKCε to generate an active kinase domain is sufficient for the implementation of these metaphase–anaphase transition controls and reveals a Caspase-7-dependent PKCε-Aurora B signalling module.

**Aurora B switches specificity to modify two sites on TopoIIα.** TopoIIα has been identified as a classical PKC substrate in previous studies[17–19] and this was determined to be a cell cycle phase specific phosphorylation of the residue S29[18]. We generated a phospho-specific antibody against this site to determine whether TopoIIα S29 could be a target of signalling through the mitosis-derived PKCε catalytic fragment. Phosphorylation of TopoIIα at S29 was observed to be most prevalent during metaphase and to a lesser extent during anaphase; however it is completely absent at cytokinesis and in non-mitotic cells (Fig. 4a). Phosphorylation of S29 was significantly reduced after PKCε inhibition with BLU577 (Fig. 4b). Strikingly, TopoIIα S29 phosphorylation was substantially more reduced after Aurora B inhibition with ZM447439 (Fig. 4b), suggestive of TopoIIα being a downstream target of the PKCε-Aurora B signalling module.

A direct relationship between Aurora B and TopoIIα has also previously been described[20]. In agreement with the observations of Coelho et al., we detect co-localisation of Aurora B and TopoIIα at the centromere in metaphase cells (Supplementary Fig. 5a). The centromeric localisation of TopoIIα was lost upon Aurora B knockdown and became more diffuse with expression of Aurora B S227A suggesting a functional relationship with the PKCε-Aurora B node. We sought to determine whether active Aurora B (both the pT232 and the pS227/pT232 phosphorylated forms) could phosphorylate TopoIIα initially using peptide arrays of the protein (Fig. 4c, d). Remarkably, the two phospho-species of Aurora B displayed distinct preferences for the phosphorylation of TopoIIα. The recombinant Aurora B wild type protein (pS227/pT232) predominantly phosphorylated TopoIIα S29 while the Aurora B S227A mutant (only phosphorylated on T232 in the activation loop) chiefly phosphorylated the T1460 site (Fig. 4c, d). This preference in substrate sites was also displayed in the phosphorylation of recombinant TopoIIα in vitro (Fig. 4e, Supplementary Fig. 5b). We further validated these as being *bona fide*, cell cycle regulated phosphorylation sites in cells by immunoblot and immunofluorescent microscopy (Supplementary Fig. 5c–f). Notably, we confirmed TopoIIα S29 as a substrate

of the PKCε-Aurora B signalling module in cells as phosphorylation at this site is significantly diminished upon expression of either the non-cleavable PKCε D383/451N or Aurora B S227A (Fig. 4f, g).

Detection of phosphorylation at TopoIIα T1460 was confounded by the fact that the phospho-T1460 antibody alone was not amenable to standard immunofluorescent staining methods, however detection by proximity ligation assay (PLA) when paired with a pan-TopoIIα antibody allowed visualisation of the localisation of T1460-phosphorylated TopoIIα. Phosphorylation of the T1460 site appears to occur in both mitotic and non-mitotic cells to varying degrees but nevertheless dependent on Aurora B as the PLA signal is diminished upon treatment with ZM447439 (Supplementary Fig. 5f). We validated the signal from the antibody pairs by peptide competition and were no longer able to detect signal in the presence of peptide with the phosphorylated-T1460 residue (Supplementary Fig. 5f).

**Aurora B S227 influences activation loop dynamics.** The profound switch in Aurora B specificity towards TopoIIα led us to examine the differences in the predicted kinase structure and the dynamics underlying the switch in Aurora B substrate specificity using molecular dynamics simulations. A 100 ns molecular dynamics simulation was carried out to explore the role of phosphorylation of Aurora B pS227 on the dynamics of the protein and the conformation of the activation loop. The simulations were carried out in the ATP-bound form and in the presence of the INCENP C-terminal domain. During the course of the MD simulation, the loop region of the Aurora B pT232 form appeared to be more flexible than the Aurora B pS227/pT232 form (Supplementary Fig. 6a). Principal component analysis (PCA) highlights that the conformation of the loop in Aurora pS227/pT232 is noticeably different from the loop in Aurora B pT232. The RMSF fluctuations of PC1, PC2, and PC3 show different dynamics in the pS227/pT232 form of Aurora B as well as a change in motion in the activation loop and INCENP (Supplementary Fig. 6b).

The dominant conformations of the pT232 and pS227/pT232 Aurora B enzymes (Supplementary Fig. 6c), extracted from the MD trajectories, were used for the molecular docking of peptides containing TopoIIα S29 and T1460 residues to assess the switch in substrate specificity observed for the pT232/pS227 Aurora B enzyme. The peptide containing TopoIIα S29 was able to bind to the groove in C-lobe of the protein structure next to the ATP-binding pocket of the dominant form of pS227/pT232 Aurora B but the monophosphorylated pT232 Aurora B dominant form could not accommodate the peptide. Similarly, the peptide containing T1460 residue docked within the binding pocket close to the ATP-binding site of pT232 Aurora B. However, this peptide occupied a much lower position and with inappropriate orientation in the pS227/pT232 Aurora B. The peptide docking was followed by 100 ns MD simulations to further explore the

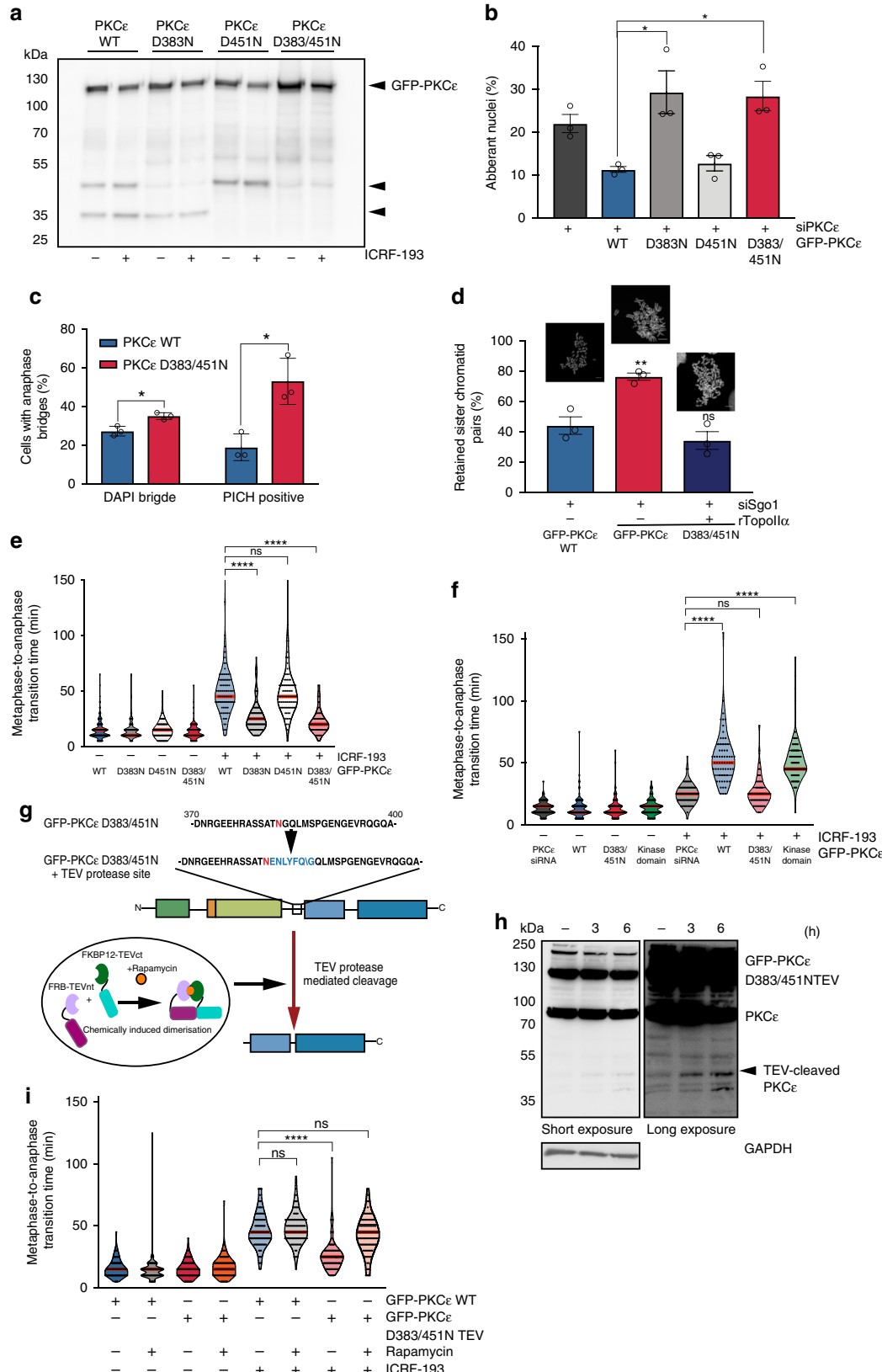

interactions between the peptides and the critical residues of the enzyme. As the catalytic mechanism of Aurora B involves acidifying the incoming OH of the substrate by aspartic acid of HRD motif, the distance between Asp200 of the motif and the OH group of TopoIIα S29 and T1460 residue was monitored

throughout the course of the simulation. The initial distance for the pT232 substrate T1460 residue was 10.768 Å—the MD simulation then pulled the substrate peptide closer to the ATP site, and the distance was reduced to 4.203 Å (Supplementary Fig. 7a, b). A similar phenomenon was observed for the pT232/

**Fig. 3 Cleaved PKCε is required for the metaphase catenation response. a** Immunoblot of the TritonX-100-insoluble fraction of DLD1 cells expressing GFP-PKCε cleavage-site mutants, after arrest with nocodazole (500 nM) ± ICRF193 (1 μM) (representative of three independent experiments).
**b** DLD1 cells expressing GFP-PKCε wild type or cleavage mutants were scored for nuclear morphology. The mean ± SD of three independent experiments is graphed (n = ≥200 cells per condition, per experiment, two-tailed Students' t test; *P = 0.023, **P = 0.008; siPKCε 22.11% ± 3.68, WT 11.4% ± 1.21, D383N 29.4% ± 8.64, D451N 12.85% ± 5.94, D383/451 N 28.53% ± 3.43). **c** Anaphase DLD1 cells expressing GFP-PKCε wild type or D383/451N were scored for DAPI-positive bridges or PICH-positive strands. The mean ± SD of three independent experiments is graphed (n = ≥20 cells per condition, per experiment) (two-tailed Students' t test, *P = 0.012, DAPI:WT 27.33% ± 2.52, D383/451N 35.06% ± 1.761, PICH: WT 19% ± 6.93, D383/451N 53.01% ± 11.88). **d** Cells scored for the percent retained (catenated) sister chromatids. The mean ± SEM of three independent experiments is graphed (n = ≥15 cells per experiment) (One-way ANOVA; **P = ≤0.0064; WT 44.24% ± 5.74, D383/451N 76.43% ± 2.36, D383/451N + rTopoIIa 34.4% ± 5.83) Scale bar = 5 μm. **e** DLD1 cells expressing GFP-PKCε wild type or cleavage mutants were treated ± 1 μM ICRF193 immediately prior to time-lapse imaging and scored for metaphase-to-anaphase transition time. Violin plots represent all cells scored. A minimum of 50 cells per condition in three independent experiments were analysed (One-way ANOVA, ****P = ≤0.0001; WT + ICRF193 46.26 ± 4.62 min, D383N + ICRF193 25.51 ± 1.32 min, D451N + ICRF193 48.9 ± 1.18 min, D383/451 N + ICRF193 23.05 ± 1.62 min). **f** DLD1 cells expressing GFP-PKCε wild type, cleavage mutants or kinase domain were treated ±1 μM ICRF193 immediately prior to time-lapse imaging and scored for metaphase-to-anaphase transition time. Violin plots represent all cells scored. A minimum of 50 cells per condition in three independent experiments were analysed. (one-way ANOVA; ****P = ≤0.0001; siPKCε + ICRF193 24.05 ± 1.15 min, WT + ICRF193 55.71 ± 2.3 min, D383/451N + ICRF193 24.92 ± 2.46 min, kinase domain 55.01 ± 2.6 min). **g** Schematic of split-TEV system. **h** Immunoblot (representative of three independent experiments) of DLD1 cells co-expressing GFP-PKCεD383/451N ± TEV recognition sequence and split-TEV (sTEVp) after ± rapamycin (100 nM) for 3 and 6 h to demonstrate TEV-dependent cleavage. **i** DLD1 cells expressing GFP-PKCε wild type or TEV-cleavage mutant were arrested with nocodazole (3 h, 500 nM) followed by ± rapamycin (2 h,100 nM). After nocodazole washout cells were treated ±1μM ICRF193 immediately prior to time-lapse imaging and scored for metaphase-to-anaphase transition time. Violin plots represent all cells scored. A minimum of 100 cells per condition in three independent experiments were analysed (one-way ANOVA; ****P = ≥0.0001; WT + ICRF193 47.15 ± 0.82 min, WT + ICRF193 + Rapamycin 47.86 ± 0.9 min, D383/451NTEV + ICRF193 25.25 ± 1.15 min, D383/451NTEV + ICRF93 + Rapamycin 44.95 ± 1.35 min).

pS227 substrate peptide TopoIIα S29, and the distance was reduced from 12.948 to 4.666 Å (Supplementary Fig. 7a, b). The post-simulation distances suggest the OH of the substrate peptide is in close proximity of the Asp200 to initiate the catalytic cycle. However, in the case of TopoIIα S29 and pT232 Aurora B, and T1460 and pS227/pT232 Aurora B, the distances, both pre-and post-simulations, remained unfavourable for phosphorylation to occur (Supplementary Fig. 7c) and the orientation of the peptides remained unfavourable.

**Aurora B phosphorylation of TopoIIα enhances decatenation**. To examine the potential function of these Aurora B sites on TopoIIα, we engineered the DLD1 cell line to express inducible GFP-TopoIIα with the S29 and T1460 residues mutated to non-phosphorylatable alanine, both individually and in combination (Supplementary Fig. 8a). Interestingly, neither site mutant nor their combination influenced the implementation of a delay to anaphase entry in the presence of ICRF193 (Fig. 5a) suggesting a function for Aurora B and the phosphorylation of TopoIIα distinct from the regulation of the SAC (see above Fig. 1g, h and ref. [2]). We next investigated whether the two distinctive forms of Aurora B would influence the activity of recombinant TopoIIα in vitro. To ensure we could observe any effect of Aurora B phosphorylation, recombinant TopoIIα was titrated to sub-optimal decatenation activity (Supplementary Fig. 8b). Aurora B WT protein (pS227/pT232) increased TopoIIα activity against catenated kinetoplast DNA, while the S227A mutant (i.e., not phosphorylated at S227) showed a reduced effect (Fig. 5b) despite both kinase preparations demonstrating comparable activity against the shared substrate, Histone H3 S10 (Supplementary Fig. 8c)[4]. This strongly suggests that phosphorylation of TopoIIα at the S29 site contributes to the resolution pathway. Further supporting this is the observation that in vivo, when decatenation is inhibited with ICRF193, phosphorylation at the S29 site persists beyond mitosis, into cytokinesis as well as being present in binucleated cells that have failed cytokinesis (Fig. 5c). To further explore the requirement for phosphorylation at these sites, we used the inducible DLD1 GFP-TopoIIα cell lines and also engineered HEK293 cells to express inducible GFP-TopoIIα WT or mutant forms with either the single or paired sites compromised: S29A, T1460A, and S29/T1460A (Supplementary Fig. 8d). We found that unlike the

ectopic WT TopoIIα, mutant forms in either of these two phosphorylation sites prevented support of the resolution pathway and led to the accumulation of catenated chromosomes (Fig. 5d, e); in cells where S29 was unable to be phosphorylated, this was commensurate with an increase in cells with aberrant nuclear morphology (Supplementary Fig. 8e). GFP-TopoIIα was extracted from the inducible HEK293 cell lines (Supplementary Fig. 8f) to examine whether the non-phosphorylatable mutant TopoIIα could still decatenate kDNA in vitro. We demonstrate a diminished decatenation ability of both TopoIIα S29 mutants (S29A and S29/T1460A) (Supplementary Fig. 8f, g) indicating phosphorylation of TopoIIα S29 enhances the decatenating ability of the enzyme. Collectively, the data indicate a bifurcation of the described Caspase7-PKCε-Aurora B pathway (Fig. 6). One arm of this pathway signals a delay in anaphase entry through PKCε-Aurora B regulation of SAC silencing in response to unresolved catenanes, whilst the other directly influences the behaviour of TopoIIα in relation to the resolution of catenation through S29 phosphorylation.

## Discussion

We have defined a mechanism by which the key mitotic kinase, Aurora B, orchestrates safe passage through the metaphase–anaphase transition in the presence of catenation. The switch in Aurora B substrate repertoire after PKCε phosphorylation of the activation loop residue S227 has profound effects on conformation of the kinase and how it interacts with substrates. This was demonstrated by the distinct difference in preference between Aurora B phoso-species for the two phosphorylation sites on TopoIIα, S29, and T1460, and we identified S29 phosphorylation as being necessary for enhanced catenation resolution. In addition, we have determined the input upstream of Aurora B which involves a non-canonical Caspase-7 signalling pathway resulting in the proteolytic cleavage of PKCε in mitosis. This effectively disengages the kinase from the well-defined mechanism of activation initiated by lipid binding at the cell membrane, potentially allowing PKCε access to the chromatin compartment and Aurora B. The data describe a novel signalling cascade engaging a genome-protective module, the PKCε-Aurora B cascade, acting both to delay metaphase in response to catenation and to promote resolution of catenation present in metaphase.

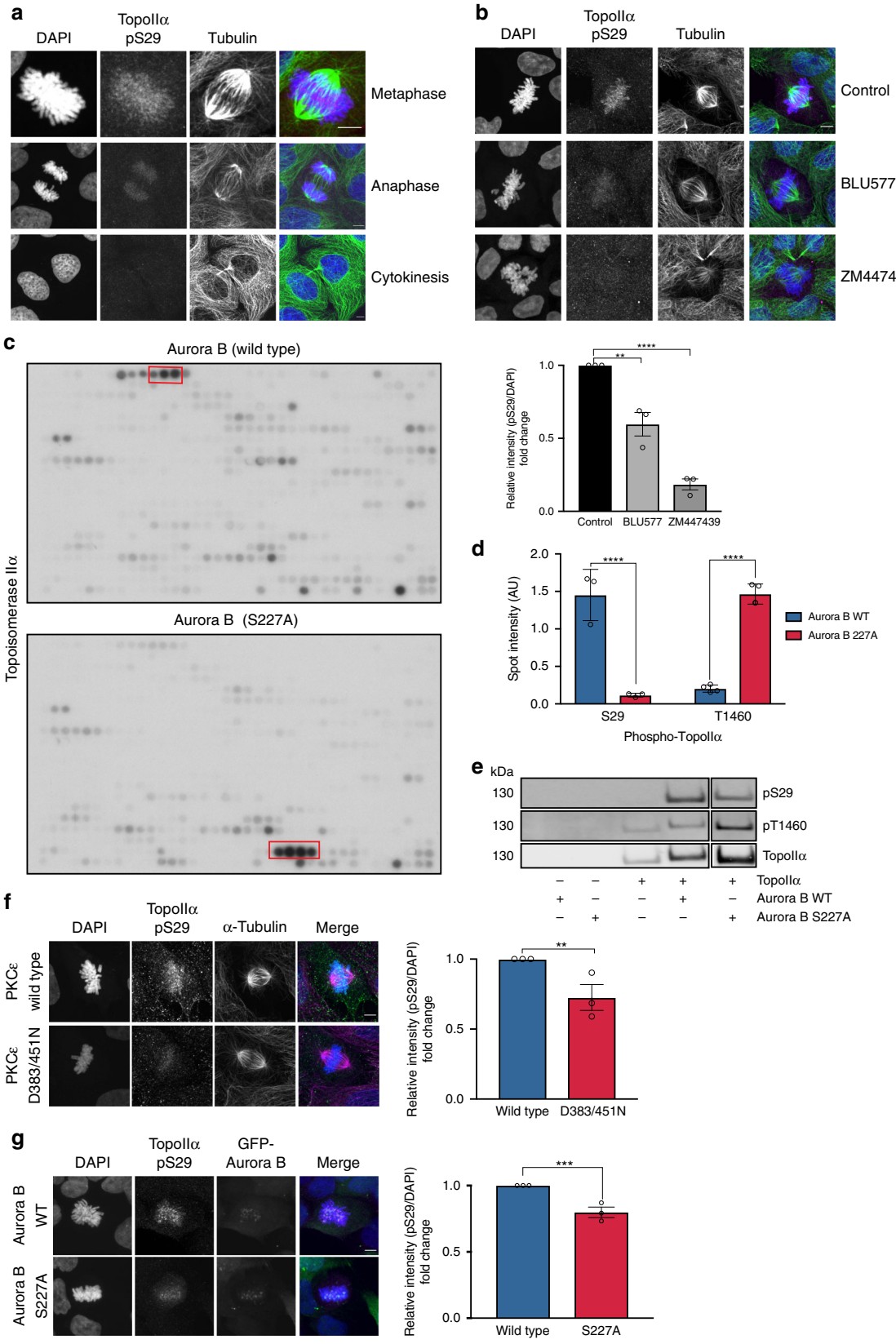

At the later abscission checkpoint, PKCε is activated by 14-3-3 binding[9] and this is associated with Aurora B phosphorylation at S227 in the activation loop, causing a switch in substrate specificity and Borealin phosphorylation at S165[4]. A protective role of PKCε at the abscission checkpoint is similarly the case at the metaphase–anaphase transition. In the presence of TopoIIα inhibition at metaphase, PKCε delays silencing of the SAC and anaphase entry by modulating dynein-mediated streaming of BubR1 away from the kinetochore[2]. This suggests that signalling through this pathway is important in both the resolution of metaphase

**Fig. 4 Aurora B switches substrate specificity to phosphorylate TopoIIα. a** Immunofluorescent images of DLD1 cells stained for TopoIIα phospho-S29 (magenta), a-tubulin (green) and DNA (blue). Images are representative of three independent experiments. Scale bar = 5 µm. **b** Representative still images of DLD1 cells stained with TopoIIα pS29 (magenta), a-tubulin (green) and DNA (blue) after 30 min treatment with the PKCε inhibitor BLU577 (500 nM) or 15 min Aurora B inhibition with ZM447439 (2 µM). Quantification of integrated pixel intensity measurements of TopoIIα pS29. Graph shows mean ± SEM of three experiments where 20 cells were scored per condition (one-way ANOVA; **$P = 0.0034$ ****$P = \leq 0.0001$; Control 1 ± 0, BLU577 0.5967 ± 0.08, ZM447439 0.18 ± 0.04) Scale bar = 5 µm. **c** Peptide array ($n = 1$) of full length TopoIIα with recombinant Aurora B wild type (top panel) or Aurora B S227A (lower panel). **d** Quantification of TopoIIα array spot intensity for S29 and T1460 peptides. Graph represents mean ± SD of the intensity of spots highlighted in red in **c** ($n = 3$ spots for S29, $n = 4$ spots for T1460). (two-tailed Students' $t$ test; ****$P = 5.17 \times 10^{-6}$(S29), $P = 1.4 \times 10^{-8}$ (T1460); S29: Aurora B WT 1.45 ± 0.34, S227A 0.12 ± 0.03, T1460: WT 0.2 ± 0.05, S227A 1.47 ± 0.13). Spot densitometry analysed by ImageJ software. **e** In vitro kinase assay of TopoIIα with Aurora B wild type or S227A kinase. TopoIIα was immunoblotted for phosphorylation at residues S29 and T1460. Immunoblot is representative of four independent experiments. **f** Representative images of DLD1 cells induced to express GFP-PKCε WT or D383/451N mutant stained for TopoIIα pS29 (magenta) and tubulin (green). Quantification of integrated pixel intensity measurements of TopoIIα pS29. Graph shows mean ± SEM of three experiments where ≥20 cells were scored per condition (two-tailed Students' $t$ test; **$P = 0.009$) Scale bar = 5 µm. **g** Representative images of DLD1 cells induced to express GFP-Aurora B WT or S227A mutant stained for TopoIIα pS29 (magenta) and GFP-Aurora B (green). Quantification of integrated pixel intensity measurements of TopoIIα pS29. Graph shows mean ± SEM of three experiments where ≥10 cells were scored per condition (two-tailed Students' $t$ test; ***$P = 0.0008$) Scale bar = 5 µm.

catenation and delayed onset of anaphase. The ability of Aurora B S227A to phenocopy PKCε mutant behaviour is consistent with previous data indicating that Aurora B S227 is a substrate of the recombinant PKCε kinase domain influencing substrate selection of Aurora B[4]. We herein provide direct evidence of PKCε phosphorylation of Aurora B S227 in mitosis, demonstrating that a PKCε-Aurora B signalling module plays a protective role at both cell cycle junctures.

We have previously reported the ability of Aurora B to switch substrate specificity in response to PKCε phosphorylation to exit from the abscission checkpoint[4]. MD simulations suggest the phosphorylation of S227 makes a significant conformational change in the loop structure and the dynamics of the protein. This change in dynamics can potentially explain the changes in the substrate specificity compared with the constitutively active Aurora B pT232. This was supported by peptide docking followed by the 100 ns MD simulation of the TopoIIα S29 peptide-bound Aurora B pS227/pT232 and the T1460 residue containing peptide-bound Aurora B pT232. The complexes remained stable throughout the course of the simulation. The bis-phosphorylated B pS227/pT232 Aurora B could not accommodate the peptide containing T1460 residue while being able to accommodate the TopoIIα S29 peptide, the reciprocal is also true, providing further insight into the substrate switch. During the MD simulations, the phosphorylated forms of Aurora B were able to pull their respective substrate peptides towards the nucleotide-binding site and close to D200 to align for catalysis. This indicates that the changes in conformation of the kinase after phosphorylation of S227 are directly related to the selection of distinct substrates.

Cleavage by Caspase-7 proteolysis in the V3 hinge domain of PKCε, resulted in a small pool of constitutively active kinase domain. Other PKC family members including PKCδ, PKCη, PKCθ are similarly cleaved by Caspase in the V3 hinge region to separate the regulatory domain and catalytically active kinase domain[21]; however these responses are pro-apoptotic signals. The outcome of Caspase-7 cleavage of PKCε appeared to be different as we observed no evidence of an apoptotic programme being triggered. In fact, Basu et al. report that Caspase-7 dependent cleavage of PKCε in MCF7 cells protects from TNF-induced apoptosis[12]. Caspase-7 has been implicated in mitotic progression; expression of the active form of the protease increases during mitosis and knockdown or inhibition of Caspase-7 in HepG2 cells prevents proliferation by arresting cells in mitosis[22]. This non-apoptotic role of Caspase-7 in mitosis is supported by a growing body of literature of alternative functions of Caspases including cell cycle regulation where, for example, Caspase cleavage of proteins such as p21[CIP1/WAF123], p27[KIP124,25], and

BubR1[26] have been reported to drive proliferation. Collectively the data indicate that PKCε is the critical target for Caspase-7 in this non-disjunction protective cell cycle pathway and that cleavage of PKCε to release the kinase domain is necessary and sufficient for the delay in the metaphase–anaphase transition as well as resolution of the catenation associated with this delay.

The mapping of the Caspase-7-PKCε-Aurora B cascade led us to test whether the catenane resolution pathway is directly influenced by the output of this cascade. TopoIIα mediates decatenation and has been identified previously as both a PKC substrate[17,18], and an Aurora B substrate in vitro[27] and a functional relationship between the two kinases and TopoIIα has been described in yeast (PKC[17], Aurora B[28]) and *Drosophila*[20]. While we cannot exclude PKCε as an upstream kinase which phosphorylates TopoIIα, inhibition with ZM447439 and expression of the Aurora B S227A mutant suggests this phosphorylation is predominantly carried out by Aurora B. PKCδ, which is also cleaved by caspases to generate a free catalytic domain, modulates TopoIIα activity in S phase in response to genotoxic stress, triggering Caspase-dependent apoptotic cell death[29]. Interestingly, Coelho et al.[20] report that Aurora B localisation and activity at the centromere is dependent on the presence of TopoII for accurate bi-orientation and chromosome segregation. Further to this, Edgerton et al.[28] demonstrated that Aurora B is recruited by the C-terminal domain of TopoIIα, independent of the enzyme's catalytic activity. We demonstrate that TopoIIα is a direct substrate of Aurora B and phosphorylation of TopoIIα enhances the decatenating ability of the enzyme required for the resolution pathway, however this interaction is independent of the catenation-associated delay to anaphase entry. Taken together, the data indicate a bifurcation of the described Caspase-7-PKCε-Aurora B pathway. One arm of this pathway signals a delay in anaphase entry through PKCε-Aurora B regulation of SAC silencing at the kinetochore in response to unresolved catenanes, whilst the other triggers phosphorylation of TopoIIα impacting catenane resolution.

In summary, the unravelling of the catenation-associated delay and resolution pathways operating through PKCε at the metaphase/anaphase transition, has revealed a series of distinctive events. There is a non-canonical, cell cycle coupled activation pathway operating through Caspase-7 resulting in PKCε cleavage. This produces a phosphorylated i.e., primed, catalytic domain which we demonstrate by ectopic expression and inducible PKCε cleavage, is sufficient to implement the delay to anaphase entry triggered by ICRF193 treatment. Having previously established that Aurora B can be a downstream target of PKCε, we demonstrate here that the ability of Aurora B to switch substrate repertoire in response to phosphorylation at S227 is important for both the delay and resolution pathways. We provide evidence that two phospho-forms of Aurora B differentially phosphorylate

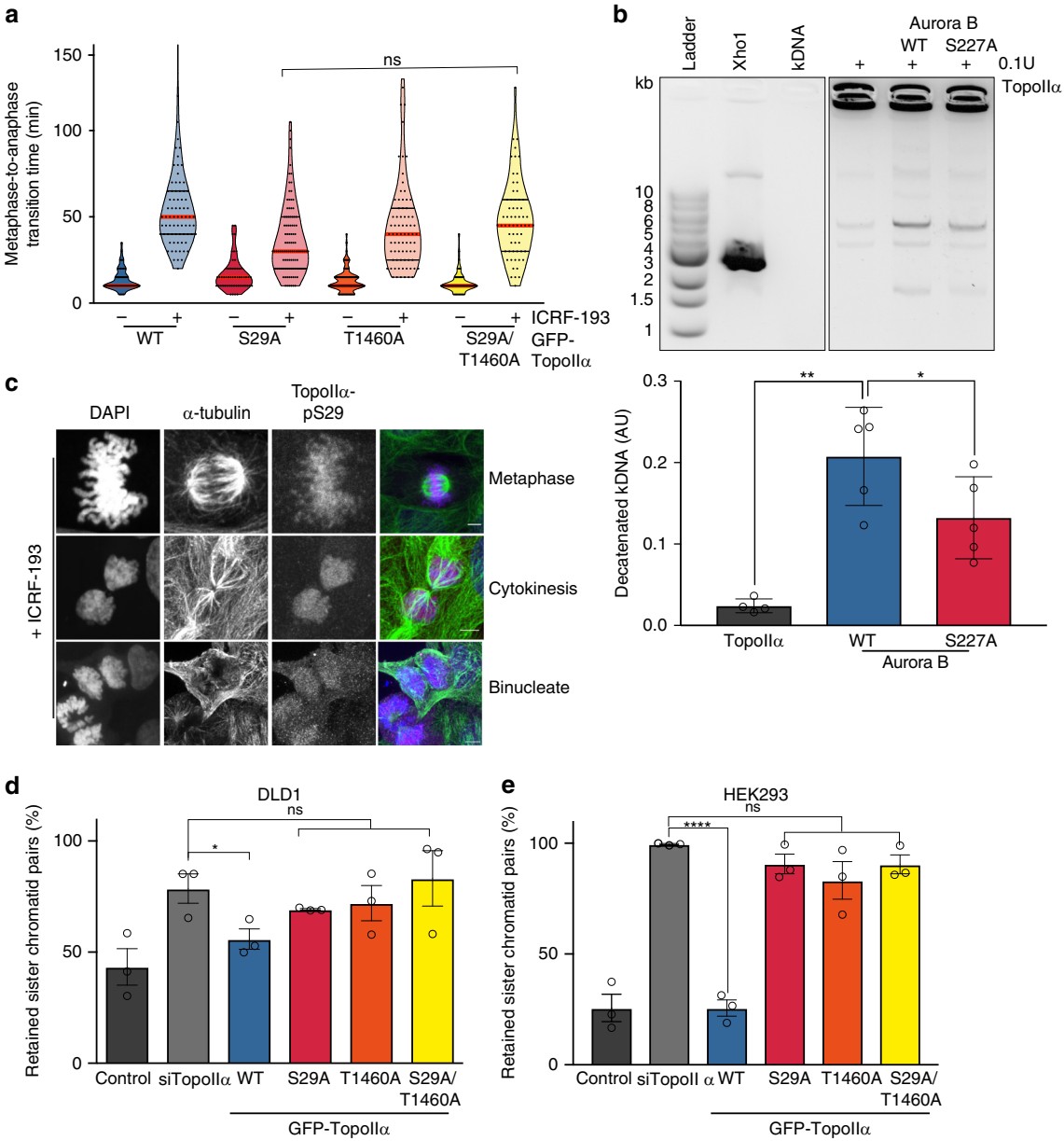

**Fig. 5 Phosphorylation is required for enhanced TopoIIα decatenation. a** DLD1 cells induced to express GFP-TopoIIa wild type or mutants were assessed for the metaphase catenation response. Cells were treated ±1 μM ICRF193 immediately prior to time-lapse imaging and scored for the time to progress from metaphase to anaphase. The mean ± SEM of three independent experiments is represented by the graph, ≥100 cells per condition were analysed. (one-way ANOVA; WT + ICRF193 55.6 ± 2.23 min S29A + ICRF193 44.18 ± 6.69 min, T1460A + ICRF193 46.11 ± 2.77 min, S29/T1460A + ICRF193 45.52 ± 1.71 min). **b** In vitro decatenation assay. Quantification of the amount of decatenated kDNA present after addition of TopoIIα and recombinant Aurora B (wild type or S227A). Graph represents the mean ± SD of five independent experiments. (two-tailed Students' *t* test, *P = 0.012, **P = 0.0025; Control 0.024 ± 0.009, WT 0.208 ± 0.06, S227A 0.132 ± 0.05). **c** Representative immunofluorescence images of DLD1 cells after 1 h treatment with ICRF193 (1 μM) showing TopoIIα pS29 (magenta), a-tubulin (green) and DNA (blue). Images are representative of three independent experiments. Scale bar = 5 μm. **d, e** Catenation spread assay analysis. **d** DLD1 cells induced to express GFP-TopoIIα wild type or mutants were scored for the per cent joined (catenated) chromatids. The mean ± SEM of three experiments where ≥15 cells per experiment were analysed is represented by the graph. (two-tailed Students' *t* test; *P = 0.047; siControl 43.38% ± 8.22, siTopoIIa 78.64% ± 6.61, WT 55.88% ± 4.58, S29A 69.24% ± 0.36, T1460A 72.07% ± 7.94, S29/T1460A 83.17% ± 12.49). **e** HEK293 cells induced to express GFP-TopoIIa wild type or mutants were scored for the per cent joined (catenated) chromatids. The mean ± SEM of three experiments where ≥15 cells per experiment were analysed is represented by the graph (two-tailed Students' *t* test; ****P = ≤0.0001; siControl 25.66% ± 6.17, siTopoIIa 99.57% ± 0.26, WT 25.62% ± 3.63, S29A 90.68% ± 4.44, T1460A 83.21% ± 8.47, S29/T1460A 90.46% ± 4.25).

TopoIIα to directly affect its activation, specifically acting in the resolution pathway (Fig. 6).

## Methods
**Reagents**. All reagents were purchased from Sigma-Aldrich unless otherwise stated. BLU577 was kindly provided by Dr Jon Roffey, Cancer Research Technology, UK.

**Cell culture**. All cell lines were obtained from ATCC and cultured in DMEM (Thermo Fisher Scientific) with 10% fetal calf serum (Thermo Fisher Scientific). Tetracycline-inducible DLD1 (a kind gift from Prof Stephen Taylor, University of Manchester) were generated using the Flp-In™ T-Rex™ system (Invitrogen) according to the manufacturer's instructions. For GFP-Aurora B, GFP-TopoIIα or GFP-PKCε expression, cells were depleted of endogenous protein by siRNA and cultured in DMEM containing 10% FCS and tetracycline (100 ng/ml) for 16 h prior

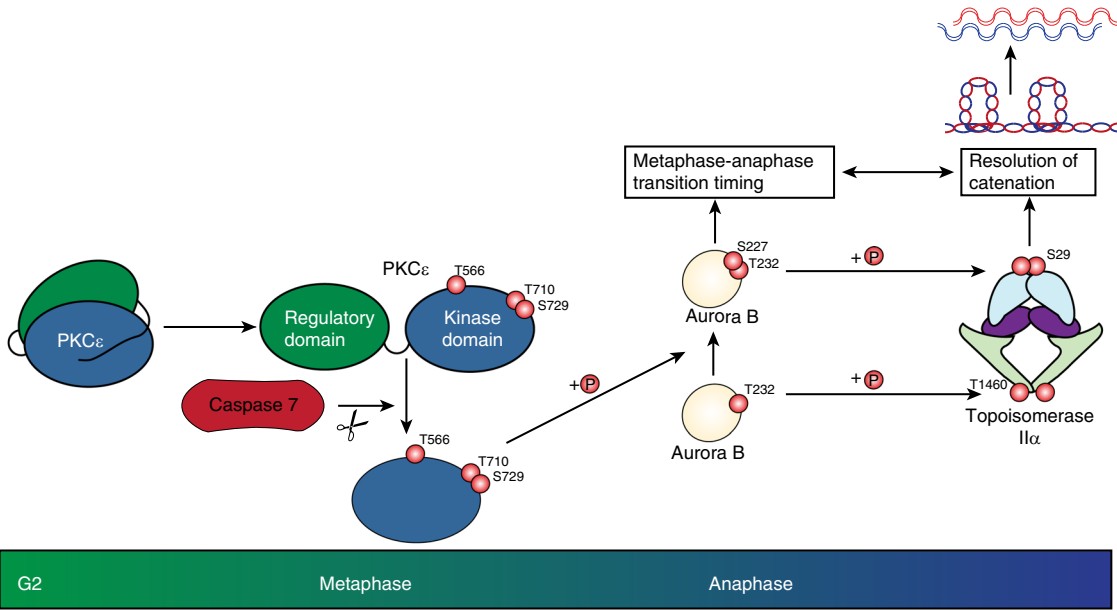

**Fig. 6 Working model of non-canonical PKCε signalling in response to excess metaphase catenation.** In mitosis, Aurora B switches substrate specificity in response to phosphorylation of S227 in the activation loop by a cell cycle-processed active fragment of PKCε. Here, we show that this switch protects from chromosome non-disjunction by delaying anaphase entry and promoting TopoIIα-dependent resolution.

to assay. Cells were treated with inhibitors for the time/s stated. Cell lines were routinely tested for mycoplasma. For cell synchronisation, cells were treated with a double thymidine block (2.5 mM), released into pre-warmed, complete media for 6 h and arrested in G2/M by adding 9 µM RO-3306 (CDK1 inhibitor) and 1 µM ICRF193. Cells were released into mitosis by washing with complete media and the indicated inhibitors and monitored until they reached mitosis.

**Plasmids.** GFP-PKCε and TopoIIα-GFP (a kind gift from Dr Christian Mielke) cDNAs were cloned into pcDNA5/FRT/TO(Invitrogen) using the InFusion cloning kit (Clontech) as per the manufacturers' instructions. For recombinant protein production, pET-Duet-1 Aurora B:INCENP was a gift from Dr Jon Elkins, University of Oxford. Site-directed mutagenesis was completed using ChangeIT mutagenesis kit (Agilent) according to the manufacturers' instructions. All clones were sequence verified.

**Primers.** See supplementary methods (Supplementary Tables 1–3) for full list of primers used in this study.

**siRNA.** For siRNA transfection, Lullaby (OZ Biosciences) was used according to manufacturers' recommendations; all siRNAs were used at a final concentration of 20 nM. The following SmartPOOLS were purchased from Dharmacon:

PKCε si1 (D-004653-01 5′-GGGCAAAGAUGAAGUAUAU-3′), Caspase-3 (Cat. L-004307 5′-CCGACAAGCUUGAAUUUAU-3′, 5′-CCACAGCACCUGG UUAUUA-3′, 5′-GAAUUGAUGCGUAGAUGUUU-3′, 5′-GCGAAUCAAUGG ACUCUGG-3′) and Caspase-7 (L-004407 5′-GGGCAAAUGCAUCAUAAUA-3′,5′ GAUCAGGGCGUGUAUUGAAG-3′,5-UACCGUCCCUCUUCAGUAA-3′, 5′-CCA GACCGGUCCUCGUUUG-3′). siTopoIIα(5′-GGUAACUCCUUGAAAGUAAA-3′, 5′-GGAGAAGAUUAUACAUGUA-3′,5′-GAUGAACUCUGCAGGCUAA-3′, 5′ CGAAAGGAAUGGUUAACUA-3′) and siAuroraB (5′- AATTAGGGATCCCT TCTTTCC-3′) were designed against the respective 3′ UTR to allow knockdown of endogenous protein with expression of ectopic GFP-tagged proteins. The remaining siControl oligonucleotides were purchased from Qiagen; cat. 1027310 (5′-AATTCT CCGAACGTGTCACGT-3′).

**Kinase assay.** Kinase assays were conducted in 50 µl reactions containing 20 mM Tris pH7.5, 5 mM MgCl2, 0.5 mM DTT, 0.2% Triton X-100, 100 µM ATP. Reactions were incubated for 30 min at 37 °C using 100ng-1 µg of each recombinant protein per reaction. Reactions were terminated by addition of NuPAGE 4× LDS sample buffer (Thermo Fisher Scientific) prior to SDS–PAGE and western blotting with the appropriate phospho antibodies to detect kinase activity against specific phospho-sites. GFP-PKCε proteins were immunoprecipitated from DLD1 cells and ATP consumption was measured by Kinase-GLO™ as per manufacturer's instructions.

**Peptide array.** 15 mers of TopoIIα with a shift of two were arrayed on nitro-cellulose by the Peptide Chemistry Laboratory of the Francis Crick Institute. Membranes were blocked in 0.2 mg per ml BSA, 20 mM Tris pH 7.5, and 0.02%

Tween-20 overnight. Membranes were subsequently incubated with the appropriate recombinant protein (Aurora B Wild type (100 µg), Aurora B S227A (100 µg)), and 10 mM MgCl₂, 100 µM[$^{32}$P]-ATP (5µCi per ml) for 10 min followed by extensive washing in H₂O and acetic acid. Membranes were then exposed to film before spot intensity analysis using the ImageQuant TL7 (GE Lifesciences).

**Chromatin fractionation.** Cells were swelled in a hypertonic buffer (10 mM Tris pH7.5, 5 mM MgCl₂, 1 mM EGTA, 1 mM DTT) for 20 min on ice before being lysed by 20 strokes through a 25 G needle. The supernatant (S1) was collected as the cytosolic fraction and the pellet was lysed in 1% Triton X-100 buffer (1% Triton X-100, 150 mM NaCl, 50 mM Tris pH7.4, protease inhibitor cocktail (Roche)) for 10 min on ice before separating the supernatant (S2) and chromatin containing pellet (P1) by high-speed centrifugation (15,000 × $g$, 20 min).

**Triton X-100 fractionation.** Cells were lysed in 5x pellet volumes of 1% Triton X-100 buffer (1% Triton X-100, 150 mM NaCl, 50 mM Tris pH7.4, protease inhibitor cocktail (Roche)) for 10 min on ice before centrifugation (13,000 × $g$, 10 min). The pellet was resuspended in 2X NuPAGE LDS sample buffer (Thermo Fisher Scientific).

**Immunofluorescence and immunoblotting.** For immunofluorescence experiments, cells were grown on 13 mm, #1.5 glass coverslips and fixed in 4% paraformaldehyde-PBS for 20 min, followed by permeabilization in 0.5% Triton X-100 for 5 min before blocking in a solution of 3% BSA/PBS. The following primary antibodies were used in these assays: mouse anti-TopoIIα (MAB4197, Millipore 1:300), rabbit anti-TopoIIα phosphoSer29 (made in-house, 1:100), rabbit anti-TopoIIα phosphoThr1460 (made in-house, 1:100), mouse anti-alpha tubulin (clone DM1A, T9026, Sigma, 1:1000), rabbit anti-PICH (H00054821-D01, Abnova, 1:300), mouse anti-Lap2B (611000, BD, 1:300), rabbit anti-Bub1 (Abcam, 1:100), rabbit anti-BubR1 (Abcam, 1:300), rabbit anti-MAD2 (Bethyl Antibodies, 1:300), human anti-centromere (Crest) (Antibodies Inc 1:300). Primary antibodies were detected with Alexa Fluor conjugated secondary antibodies diluted 1:1000 in 3% BSA/PBS (Goat anti-mouse AlexaFluor 488—A11001; Goat anti-rabbit AlexaFluor 488—A11008; Goat anti-mouse AlexaFluor 555—A21422; Goat anti-rabbit Alex-aFluor 555 Conjugated secondary antibody—A21428; Goat anti-mouse AlexaFluor 647—A32728; Goat anti-rabbit AlexaFluor 647—A21244; Goat anti-human Alex-aFluor 647—A21445, Life technologies). All coverslips were mounted using Pro-Long Diamond with DAPI (Invitrogen).

For immunoblotting, cells were lysed in 1X NuPAGE LDS sample buffer (Thermo Fisher Scientific) and sonicated for 3 × 10 s on ice. Proteins were separated by SDS–PAGE and transferred to PVDF membranes (Millipore). The following primary antibodies were used: mouse anti-Aurora B (AIM-1,611082, BD, 1:2000), rabbit anti-Aurora B phosphoThr232 (TA319253, Origene, 1:1000), rabbit anti-Aurora B phosphoSer227 (made in-house, 1:1000), mouse anti-alpha tubulin (clone DM1A, T9026, Sigma, 1:10000), mouse anti-TopoIIα (clone KiS1, MAB4197, Millipore 1:5000), rabbit anti-TopoIIα phosphoSer29 (made in-house, 1:1000), rabbit anti-TopoIIα phosphoThr1460 (made in-house, 1:1000), rabbit anti-Histone H3 (9715, Cell Signaling, 1:5000), rabbit anti-Histone H3

phosphoSer10 (9701, Cell Signaling, 1:2000), rabbit anti-PKCε (sc-214, Santa Cruz, 1:1000), rabbit anti-PKCε phosphoSer729 (ab63387, Abcam 1:1000), rabbit anti-PKCε phosphoThr566 (made in-house, 1:500), rabbit anti-PKCε phosphoThr710 (made in-house, 1:500), mouse anti-GFP (made in-house, 1:1000), rabbit anti-Caspase-3 (AB1899, Millipore, 1:500), mouse anti-Caspase-7 (551238, BC Biosciences, 1:1000), mouse anti-GAPDH (clone 6C5, MAB374, Millipore, 1:5000), anti-cleaved PARP Asp241 (9541, Cell Signaling, 1:500). Mouse and rabbit HRP-conjugated secondary antibodies (NA931 and NA934, GE Lifesciences) were used at dilutions between 1:5000 and 1:10000. Chemiluminescence was detected using Lumanata Classico western HRP substrate (Millipore) and imaged using the ImageQuant 4000 mini (GE Lifesciences). Band densitometry was carried out using ImageJ software and normalised to a loading control. Uncropped western blots are provided as Supplementary Fig. 9.

**Proximity ligation assay.** Cells were grown in eight-well chambered slides (Falcon) and fixed as for immunofluorescent imaging. PLA was conducted using a kit (Sigma-Aldrich) as per manufacturers instruction using the antibody pairs: anti-GFP (rabbit)(Abcam, ab290), anti-TopoIIα (mouse)(Millipore) and anti-TopoIIα, anti-TopoIIα phospho-T1460 (rabbit)(made in-house). Non-specific IgG (rabbit sc-2027 and mouse sc-2025, Santa Cruz) were used in conjunction with anti-TopoIIα and anti-TopoIIα phospho-T1460 (respectively) as negative controls to demonstrate the specificity of the PLA reaction.

**Microscopy.** For live-cell time-lapse microscopy, cells were cultured on LabTek chambered coverglass slides (Nunc) in Leibovitz $CO_2$-independent media (Thermo Fisher Scientific). A low light level inverted microscopy (Nikon TE2000) imaging system equipped with a laminar-flow heater to maintain a constant temperature of $37 \pm 0.001$ °C, a PlanFluor 40′ DIC lens and a Xenon lamp for fluorescent excitation. Images were taken using a high quantum efficiency CCD camera (Andor Ixon) every 5 min using MetaMorph (Version 6.3) software. Still images were taken using an inverted laser scanning confocal microscope (Carl Zeiss LSM780) equipped with a 63′ Plan-APOCHROMAT DIC oil-immersion objective using ZEN (Version 2.3 SP1) software. Images were prepared for publication using ImageJ software.

**Catenation spread assay.** For measurement of metaphase catenation, cells were treated with siSgo1 for 24 h, followed by 1 h treatment with nocodazole to collapse the mitotic spindle, to aid spreading. Cells were collected by shaking off the mitotic cells and resuspended in a hypotonic solution of 75 mM KCl and incubated at 37 °C for 30 min to expand the cell. Cells were then resuspended in 3:1 methanol:acetic acid and fixed overnight at −20 °C. Cells were then spread onto glass slides by dropping from 1 m height. For assays where TopoIIα was reintroduced, recombinant TopoIIα (1 U/μl, TopoGen) was incubated in the hypotonic step where the cell membrane becomes hyperpermeable. The hypotonic buffer used here contained 5 mM Tris-Cl, pH 8.0, 75 mM KCl, 10 mM $MgCl_2$, 0.5 mM ATP, 0.5 mM dithiothreitol. We confirmed by video microscopy that there was no significant difference between the time that all samples had been arrested in mitosis at the start of the assay. Images processed using ImageJ software, and scored manually after blinding.

**Immunofluorescent intensity profiles.** TopoIIα pS29 signal intensity was quantified using a custom-built script and the commercial software package MATLAB (MATLAB 2016a, MathWorks). The background pS29 signal for each cell was subtracted and the mitotic chromosomes were segmented by Otsu's method of thresholding of the DAPI channel. Subsequently, the pS29 signal intensity was normalized to the DAPI signal intensity and DAPI area to account for potential differences in chromatin condensation. At least ten images were analysed and the mean and S.E.M. of at least three independent experiments was quantified.

**Expression and purification of recombinant protein.** Recombinant Aurora B (WT and S227A) and PKCε-kinase domain were expressed and purified as previously described[4]. Aurora B WT and S227A proteins were assessed for their phosphorylation state (pS227 and pT232) and kinase activity against the substrate Histone H3 S10 prior to use in all assays. Upon highly efficient production (i.e., high concentration) both sites become phosphorylated in vitro.

**Kinetoplast DNA decatenation assay.** For in vitro assessment of TopoIIα decatenation efficiency, the TopoGEN kinetoplast DNA (kDNA) assay was utilised with the following deviations to the manufacturers' instructions. Recombinant TopoIIα was incubated with recombinant Aurora B WT (pS227/pT232) or Aurora B S227A (pT232) at 30 °C for 30 min. Reactions contained 0.1 U recombinant TopoIIα, 50 ng of recombinant Aurora B, 100 nM ATP, and 1X reaction buffer (Topogen) made up to 20 μl with kinase dilution buffer. Reactions were terminated by addition of DNA gel loading dye (Thermo Fisher). kDNA was separated by electrophoresis on 1% agarose gel containing GelRed™ fluorescent DNA stain. Images were captured on UVP BioDoc-It2 imaging system using VisionWorks analysis software (Analytik Jena), and the kDNA decatenation quantified by calculating band densitometry using ImageJ analysis software.

**TopoIIα extraction and decatenation assay.** HEK293 cells were induced to express GFP-TopoIIα for 24 h while endogenous TopoIIα was depleted with siRNA. Cells were twice pelleted and resuspended in ice-cold TEMP buffer (10 mM Tris pH7.5, 1 mM EDTA, 4 mM $MgCl_2$, 0.5 mM PMSF) before being left on ice for 10 min. Lysates were homogenised by ten strokes through a 25 G needle before nuclei were pelleted by centrifugation ($1500 \times g$, 10 min). The nuclear pellet was resuspended in TEMP buffer and washed, then resuspended in 4 pellet volumes of TEP buffer (10 mM Tris pH7.5, 1 mM EDTA, 0.5 mM PMSF) and an equal volume of 1 M NaCl added. Preparations were vortexed and left on ice for 60 min. Finally, nuclear preparations were centrifuged at 4 °C for 15 min at $15,000 \times g$. Supernatants were diluted with 4X NuPAGE LDS sample buffer and immunoblotted for TopoIIα levels. TopoIIα levels were equalised prior to use in the TopoGEN kinetoplast DNA assay as above.

**Recapitulating PKCε cleavage.** To assess the effect of recapitulating PKCε cleavage on the metaphase–anaphase transition time, DLD1 GFP-PKCε TEV D383/451 N expressing pmCherry-H2B cells were transfected with siPKCε, after 24 h doxycycline (100 ng/ml) was added and incubated at 37 °C and 10% $CO_2$ for 16 h. Cells were incubated for 3 h with 500 nM nocodazole to arrest the cells in prometaphase, followed by a further 2-h incubation with 100 ng rapamycin/ml to drive chemically induced dimerisation of TEV, and, as a result, PKCε cleavage. Media were aspirated and cells were washed three times with pre-warmed media to release the prometaphase arrest. Media were then replaced with 500 μl Leibovitz $CO_2$-independent media (Gibco) ± 1 μM ICRF193. Live-cell time-lapse images were obtained as above.

**Homology modelling.** The enzyme structure was obtained by homology modelling using SwissModel web server with the same amino acid sequence with two different templates of 4AF3 with 99.69% sequence identity and $5 \times 3$ F with 31.91% sequence identity to the target amino acid sequence with Uniprot FASTA code Q96GD4. The obtained model using 4AF3 as the template excludes the N-terminal and the second model was applied to amend the structure of the enzyme. The generated enzyme structure was considered in the presence of IN-box segment of INCENP, ATP, and $MG^{2+}$ ions in two separate states of mono- and bis-phosphorylated forms by phosphorylation of Thr232 and Ser277 residues located in the loop (residues 220–250).

**Molecular dynamics simulations for dominant conformations.** A total of 100-ns molecular dynamics simulations by AMBER 16 were performed for each system after minimising them with and without restraint. This was done in two stages with 2500 cycles of steepest descent followed by 2500 cycles of conjugate gradient minimisation and heating up to 310 K within 500 ps and equilibrating for 1 ns using. The cutoff radius of non-covalent interactions was set to 12 Å. Also, periodic boundary conditions, particle-mesh Ewald method, to calculate the infinite electrostatics without truncating the parameters, and SHAKE algorithm, to consider all bonds in which the hydrogen atom was present in a fixed state and the others in their equilibrium values, were employed. The force fields parameters for the phosphorylated amino acids were generated using the ANTECHAMBER module of the AMBER program. Ten dominant conformation structures after downsampling of the obtained MD trajectories were extracted using MDAnalysis to run molecular docking.

**Peptide–protein molecular docking.** To complement the experimental characterisation of the protein–peptide complexes, we determined the peptide binding site in the protein and mode of interaction between the peptides and their corresponding protein kinase in the mono- and bis-phosphorylated states of Aurora B kinase using HPEPDOCK web server and Pep-SiteFinder service to run peptide–protein molecular docking and finding the key residues respectively. The peptide containing TopoIIα T1460 residue (QKPDPAKTKNRRKRKP) and TopoIIα S29 peptide (NEDAKKRLSVERIYQKK) were considered as the peptide substrates for monophosphorylated (pT232) and bis-phosphorylated (pS227/pT232) forms of the enzyme, respectively.

**Molecular dynamics simulations of protein–peptide complexes.** A total of 100 ns molecular dynamics simulations by AMBER 16 were performed for the best poses of the peptides, located in the localised binding site by the above approaches, in complex with the enzymes. The parameters were considered as mentioned in the first round of MD simulations performed to extract the dominant conformations of mono and bis-phosphorylated Aurora B kinase.

**Principal component analysis (PCA).** PCA has been used to investigate the correlations among various important regions of the complexes in the course of MD simulations and to distinguish functional motions from the thermal and accidental motions. The fluctuations of the residues in critical regions were monitored using three different modes of PCA (PC1, PC2, and PC3).

**Statistical analysis.** For experiments where the data includes more than two conditions, a one-way ANOVA using multiple comparisons was used, in all other cases an unpaired t test was used for analysis. Prism software (Graphpad) was used for all calculations. The level of statistical significance is represented as follows: ns = $P > 0.05$, *$P \leq 0.05$, **$P \leq 0.01$, ***$P \leq 0.001$ and ****$P \leq 0.0001$.

**Reporting summary**. Further information on research design is available in the Nature Research Reporting Summary linked to this article.

## Data availability

The data that support the findings of this study are available from the corresponding author upon reasonable request.

## Code availability

Custom code developed for the immunofluorescence intensity profiling is available upon request.

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

## Acknowledgements

This work was supported by the Francis Crick Institute, which receives its core funding from Cancer Research UK (FC001130), the UK Medical Research Council (FC001130) and the Wellcome Trust (FC001130). We thank all of the Francis Crick Institute core facilities for valuable support throughout this project, in particular, the light microscopy, peptide chemistry and protein production facilities. We thank Dr Mathias Cobbaut for helpful discussions and Dr Jeremy Carlton for critically reading the manuscript.

## Author contributions

T.N.S., J.R.K., N.B., K.M.R., F.F., and P.J.P. devised experiments, J.R.K., T.N.S., S.J., S.M., and N.L. carried out experiments. D.J. and S.F. synthesised peptides and arrays, S.K. prepared and purified recombinant proteins. K.M.R., T.N.S., and P.J.P. wrote the paper.

## Competing interests

The authors declare no competing interests.
