## [Peer Review File · Nature Communications]

Reviewers' comments:

Reviewer #1 (Remarks to the Author):

Parker and colleagues have shown before that in transformed cells with compromised G2 checkpoint, PKC ϵ regulates the exit from the spindle assembly checkpoint (SAC) when challenged with catenation stress. PKC ϵ was also shown to regulate exit from abscission checkpoint by phosphorylating Aurora B at S227 and altering its substrate specificity. The manuscript by Kelly et al. now examined what triggers PKC ϵ mediated metaphase delay and catenane resolution. Basu et al. previously showed that PKC ϵ is a substrate for caspase-7, which cleaves PKC ϵ after D393 and D451 during apoptosis, and the fragment generated by cleavage of PKC ϵ after D393 is catalytically active and was associated with its anti-apoptotic function. In the present manuscript, the authors showed that PKC ϵ is also cleaved by caspase-7 in cells arrested in metaphase and this cleavage is important for catenation-associated delay in the metaphase/anaphase transition and catenane resolution. This observation is novel since it provides evidence for non-apoptotic function of caspase-7. Moreover, the authors demonstrated that the switch in substrate specificity of Aurora B by PKC ϵ leads to phosphorylation of topoisomerase IIa (TopoIIa) at S29, which is important for catenane resolution whereas phosphorylation of TopoIIa by Aurora B at both S29 and T1460 is needed for completion of decatenation. Thus, the caspase-7-PKC ϵ -Aurora B-TopoIIa pathway plays an important role in maintaining chromosome integrity. These observations are important, and should be of interests to the readers of Nature Communications. There are, however, several comments that need to be addressed.

Suppl. Fig. 1A: It appears that treatment of DLD1 cells with nocodazole decreased the mobility of full-length PKC ϵ . Perhaps, the cleaved fragment of PKC ϵ causes phosphorylation of PKC ϵ .

Suppl. Fig. B vs C: It is not clear why endogenous full-length PKC ϵ in DLD1 cells is much lower compared to HeLa cells whereas ectopically expressed full-length PKC ϵ in HeLa cells is much lower compared to DLD1 cells.

Suppl. Fig. 1C: PKC ϵ cleavage fragments could be detected in U2OS cells even without nocodazole treatment.

Suppl. Fig. 1D & Fig. 2E: The quality of the immunoblots is poor.

Suppl. Fig. 1H: Nocodazole appears to increase both the full-length and cleaved fragments of PKC ϵ .

Fig. 3A and line 102-107: Authors mentioned that cleavage of PKC ϵ to the free kinase domain was blocked by caspase-7 knockdown. However, caspase-7 siRNA had only a modest effect on GFP-PKC ϵ

processing. Basu et al. have shown before that PKC ϵ cleavage was inhibited by caspase-3/-7 inhibitor in MCF-7 cells, which lack caspase-3.

Suppl. Fig. 2C, 2D & 2E: How is caspase-7 activated by nocodazole? Activation of effector caspases, such as caspase-7, requires processing by apical caspases, which are activated by apoptotic stimuli. Taxol, which stabilizes microtubules and inhibits microtubule detachments, can also induce apoptosis. Perhaps, stress caused by nocodazole treatment can cause low level of caspase-7 activation, which is sufficient for SAC regulation but not induction of apoptotic cell death. This is consistent with the slight increase in cleaved PARP by nocodazole treatment as seen in Suppl. Figure 2C.

Fig. 3E, right panel: The resolution of the catenated chromatids in GFP-Aurora B S227A transfected cells is low.

Although the authors have shown that Aurora B S227A mutant phenocopied the PKC ϵ caspase cleavage mutants, there was no direct demonstration that cleavage of PKC ϵ was responsible for Aurora B phosphorylation at Ser227 in vivo. The authors, however, demonstrated that phosphorylation of TopoII α at S29 is diminished in cells expressing non-cleavable PKC ϵ in Suppl. Figure 3F.

Fig. 5A: The authors should provide more details regarding how decatenated kDNA was quantified. Based on the DNA gel, there does not appear to have much difference in the dectenated DNA in the presence of 0.1 and 1.0 μ g WT Aurora B or 0.01 or 0.1 μ g S227A mutant Aurora B. Does non-cleavable mutant PKC ϵ affect decatenation?

Minor comments:

Pg. 10, Line 449 and 481: Reference 2 and 15 are same.

Pg. 3, Line 146: There is an extra bracket at the end.

Pg. 11, Line 534-537: The sentence should be corrected.

Pg. 25: "Supplementary Figure 5" should be "Supplementary Figure 4."

Reviewer #2 (Remarks to the Author):

DNA catenation needs to be resolved properly during cell division to avoid genome instability. How DNA catenation resolution is coordinated during mitotic progression is not completely understood. In the current study, Kelly et al. provide some evidence to suggest the involvement of a caspase-triggered kinase cascade in this process. Although the study contains several interesting observations, key conclusions are not sufficiently supported by actual data. Furthermore, the technical quality of many experiments in the study is low. These deficiencies cast doubt on the overall validity of their conclusions.

Major deficiencies

(1) No experiments in the current study link PKC ϵ cleavage to Aurora B phosphorylation at S227. The authors should examine whether Aurora B pS227 signals are diminished in cells expressing non-cleavable forms of PKC ϵ or depleted of caspase 7.

(2) The catenation assay used in the paper is indirect. They have used paired chromatids in Sgo1-depleted cells as a measure of decatenation failure. While reasonable, other alternative explanations exist. For example, a defect in cohesin-removal pathway would also produce more paired chromatids in Sgo1-depleted cells. This is especially important given the roles of Aurora B in cohesin removal. The authors should examine whether cohesin is indeed removed in their catenation assays. In addition, they should stain the spreads with PICH, a marker of the expected ultrafine threads connecting chromatids that are not completely decatenated.

(3) In Figure 2E, why are there so many GFP-PKC ϵ degradation species even in the absence of TEV or rapamycin? I found it surprising that the induction of a cleavage product of PKC ϵ among the sea of similar cleavage products would produce any predictable cellular phenotypes as shown in Figure 2F.

(4) The effects of Caspase 7 depletion on the cleavage of GFP-PKC ϵ are marginal. Clearly other proteases are involved. Also, can the cellular effects of Caspase 7 depletion be rescued by the expression of an RNAi-resistant transgene?

(5) Results in Figure 4B are flawed. The total TopoII α blot shows uneven loading in the reactions. This needs to be corrected. It appears that the pT1460 antibody can recognize unmodified protein.

(6) In Figure 5A, the in vitro decatenation assay is not convincing. Only very small amount of the catenated DNA was decatenated in the assay. The band is barely visible. In addition, TopoII α S29A and T1469A mutants need to be included in the assay as additional controls.

(7) The specificity of the TopoII α pS29 antibody needs to be further verified by staining cells depleted of endogenous TopoII α and expressing S29A. These cells are available as they were used in Figure 5C.

(8) The effects of TopoII α on catenation (even with the caveat mentioned in point 2) are not compelling. Marginal differences are observed even with artificial conditions (i.e. Sgo1 depletion).

Reviewer #3 (Remarks to the Author):

The authors report that DNA catenation in early mitosis triggers a cascade of events. In the pathway, Caspase-7 cleaves and activates PKC ϵ , which in turn acts on Aurora B to both bind topoisomerase IIa and induce its phosphorylation, the purpose being to delay mitotic exit and enable DNA strands to decatenate.

Comments

Overall, the manuscript should be able to demonstrate that Caspase-7 is a key to topoisomerase IIa induction of decatenation in mitosis. This could be an interesting contribution to the field, but there are many serious problems with the manuscript that prohibit publication. In general, data and images are substantially deficient, as detailed below. Further, the paper has been constructed in many places of data that are unacceptably sloppy and unconvincing. These deficiencies must be corrected.

The text does not indicate why the native PKC ϵ has to be siRNA suppressed and replaced by GFP tagged PKC ϵ in every case where this has been done. Supplemental Figure 1a shows that the native PKC ϵ is strongly detectable by western blot, so it is not clear why the native form was not subjected to western blot in Figure 1, or used for western blot without siRNA knockdown. GFP-PKC ϵ is obviously necessary for mutant analysis, but its use elsewhere remains to be explained. The authors state in Methods that GFP expression always entails native protein knockdown. It appears, however, that PKC ϵ knockdown and replacement with GFP-PKC ϵ is not always done. For example, Supplementary Figure 1K shows GFP-PKC ϵ (and mutant) expression with or without knockdown of the WT. This all needs clarification and explanation.

Figure 1a. GFP-PKC ϵ +0.1% Triton X extraction yields a centrosomal signal (this is unmistakable, and a double label would confirm it). This is a major problem with respect to interpretation of the overall message of the manuscript. The text states, and logic dictates, that the signal is chromosomal. The images show otherwise. A further major problem is that the GFP tag is N-terminal, identifying the location of the regulatory domain after caspase cleavage, while the interest is in where the catalytic domain localizes. This result is key to understanding the overall results of the manuscript, and is unacceptable.

Figure 1b. The TX-100 insoluble fraction is probed by western blot, but Methods neither describes the separation step, nor the TX-100 extraction procedure. Histone H3, a highly abundant protein, should not be such a light band on westerns in this figure. This indicates that there was a lack of attention in construction of the manuscript. There is no such problem in Suppl Fig 1.

Figure 1d. Molecular mass markers should mark the full range of the proteins separated. This is not acceptable. The markers should have been the same markers used in Fig 1b.

Figure 2c. The large error bars suggest a variable that is not accounted for between experiments. Without the raw data one can only hope a computer derived P value gives sufficient confidence in this t-test to warrant the conclusion there is no meaningful difference. The values used for calculation should be the means of the three independent experiments.

Did the authors use $n=3$ or $n=45$ for calculation? If $n=3$, the variance is too great to conclude much. If $n=45$, the calculation was done incorrectly (samples are not means). The error bars for identical sample sizes are far cleaner in Figure 3e. This indicates the authors can do the experiment correctly and this should be redone.

Figure 2e. The TEV-cleavage mutant experiment was particularly convincing in Fig 2f, but the western blot that underlies the experiment was unacceptably noisy and the TEV cleaved product was a rather weak band. Other similarly probed blots were clean. Surely the authors have a cleaner experiment to present as several experiments were run. The authors must replace this blot.

Figure 3a. Is this total cell extract or the insoluble fraction? Not stated.

Figure 4. Overall an impressive and convincing figure, except for the inexplicable lack of stain for the spindle in part d second row. The peptide array experiment was surprising and the kinase assay was convincing.

Figure 5a. The decatenated bands are extremely difficult to see, and therefore it is difficult to agree with the quantitation presented here. An ideal decatenation assay with highly visible decatenated

bands can be found, for example, in Wang L, et al. J Cell Sci 123, 806 (2010), Figure 3C. This figure should look like that example.

Suppl Fig 1a. The blot could have been run and probed for full-length and fragments of the triton insoluble fraction. Why is the full length only shown for the soluble fraction?

Suppl Fig 1f. It would be reassuring to assay the reaction rates in the kinase assays, to determine that different constructs have the same catalytic efficiency, and that the mutants are truly like the WT. As it is, one cannot conclude that catalysis is completely unaffected by mutation.

Suppl Fig 2b. Both ICRF-193 and nocodazole are toxic to cells on long exposure. It is odd that viability over 24 hr is not influenced by exposure to these drugs. Explain.

Suppl Fig 2i. The absence of CREST signal in the lower two plates is inexplicable (left images). The result +ICRF-193, +nocodazole is not credible (right images). The cell chosen appears to be nearly aligned at metaphase, which is not possible at the high concentration of nocodazole used. Normally one would find a pro-metaphase array, as shown above in this figure part. Must be explained or redone. DAPI imaging is too variable throughout the figure, when it should be uniformly bright. Normally this should be a control signal that indicates microscope and camera settings were correct.

Suppl Fig 3a. The siNTC experiment result is simply wrong. There is a bright green signal in the color channel on the right, that indicates the presence of GFP-AurB, while the GFP-AurB channel shows no signal. Either GFP-AurB was expressed here and the color channel is correct, or GFP-AurB was not expressed, and the black and white channel is correct. The left-hand legend does not indicate GFP-AurB is present, but the label above the third lane indicates it is present, while the figure legend suggests it is absent. What is going on?

In any case, evidence for overlap at the centromere is scant. An overlap signal between red and green channels should be clearly yellow, and should be evident in the first row, but is only partly evident in the third row.

It is fortunate that others have found this centromeric association of Aurora B and topoll, particularly the Coelho et al paper (ref 21) cited by the authors. The Coelho paper should stand as a model to the authors of what excellent mitotic chromatin microscopic images should look like (compared to this manuscript's figure 3e for example, which again shows highly variable stain and does not give confidence that accurate quantitation can be done).

Suppl Fig3f and g. The authors use quantification of pixel intensity to graph their result. But given the high variability in microscopic image intensity presented elsewhere (remarks on Suppl Fig 2i for example), the authors must measure pixel intensity relative to another stain intensity, such as CREST or DAPI, which remains constant from cell to cell. Methods states that DAPI was used to define the

area of localization for the other channel, but gave no indication it was used as a signal quantitation control.

Suppl Figs 1i, 2g. The control chromosome bridging is pretty high, suggesting that a fraction of the cells do not adequately decatenate, no matter what. This is surprising, as one function of the delay to anaphase checkpoint is to assure complete decatenation.

Suppl Fig 4d has very high control values for bridging, approaching the level of mutant bridging in Suppl Fig 1l. The experiment should be redone. As stated above, the function of the delay to anaphase checkpoint is to assure complete decatenation, in which case bridging (except ultrafine bridges that resolve) should be minimal. The critical diagnostic criterion should be failure of abscission if meaningful bridging persists, and cells should instead be scored for abscission (again, see Wang L, J Cell Sci 123, 806 (2010), Figure 2). Measuring abscission failure, WT cells should be negative.

Minor points:

The authors do not mention that ICRF-193 is a topoisomerase inhibitor, nor that nocodazole is a microtubule assembly inhibitor that induces mitotic arrest, nor that RO-3306 is a cyclin dependent kinase inhibitor, information that would be useful to those not expert in the field.

The Figure 2 legend title is a sentence fragment, and the first sentence in the legend is garbled (adn).

Suppl Fig 4 is mislabeled "Suppl Fig 5".

The authors vaguely state that recruitment of AurB by TopoII is "mediated by the C-terminal domain" (line 175). It is the C-terminal domain of TopoII that recruits.

Reviewers' comments:

Reviewer #1 (Remarks to the Author):

Parker and colleagues have shown before that in transformed cells with compromised G2 checkpoint, PKC ϵ regulates the exit from the spindle assembly checkpoint (SAC) when challenged with catenation stress. PKC ϵ was also shown to regulate exit from abscission checkpoint by phosphorylating Aurora B at S227 and altering its substrate specificity. The manuscript by Kelly et al. now examined what triggers PKC ϵ mediated metaphase delay and catenane resolution. Basu et al. previously showed that PKC ϵ is a substrate for caspase-7, which cleaves PKC ϵ after D393 and D451 during apoptosis, and the fragment generated by cleavage of PKC ϵ after D393 is catalytically active and was associated with its anti-apoptotic function. In the present manuscript, the authors showed that PKC ϵ is also cleaved by caspase-7 in cells arrested in metaphase and this cleavage is important for catenation-associated delay in the metaphase/anaphase transition and catenane resolution. This observation is novel

since it provides evidence for non-apoptotic function of caspase-7. Moreover, the authors demonstrated that the switch in substrate specificity of Aurora B by PKC ϵ leads to phosphorylation of topoisomerase IIa (Topolla) at S29, which is important for catenane resolution whereas phosphorylation of Topolla by Aurora B at both S29 and T1460 is needed for completion of decatenation. Thus, the caspase-7-PKC ϵ -Aurora B-Topolla pathway plays an important role in maintaining chromosome integrity. These observations are important, and should be of interests to the readers of Nature Communications. There are, however, several comments that need to be addressed.

Suppl. Fig. 1A: It appears that treatment of DLD1 cells with nocodazole decreased the mobility of full-length PKC ϵ . Perhaps, the cleaved fragment of PKC ϵ causes phosphorylation of PKC ϵ .

Comment added in text (line 82): PKC ϵ has been previously reported to be phosphorylated during mitosis at S346, S350, S368 to facilitate 14-3-3 binding (Saurin *et al.* NCB 2008), resulting in slower migration of full length PKC ϵ bands by western blot of the TritonX-100 soluble fraction (Figure S1A).

Suppl. Fig. B vs C: It is not clear why endogenous full-length PKC ϵ in DLD1 cells is much lower compared to HeLa cells whereas ectopically expressed full-length PKC ϵ in HeLa cells is much lower compared to DLD1 cells.

Figure panels have now been labelled correctly.

Suppl. Fig. 1C: PKC ϵ cleavage fragments could be detected in U2OS cells even without nocodazole treatment.

Comment added in text (line 94): Some cleavage of PKC ϵ is evident in the asynchronous cells and we attribute this observation to these cells being in logarithmic growth phase therefore cultures would contain cells in each phase of the cell cycle, including mitosis.

Suppl. Fig. 1D & Fig. 2E: The quality of the immunoblots is poor.

Supplementary figure 1D and Figure 2E have been improved and replaced.

Suppl. Fig. 1H: Nocodazole appears to increase both the full-length and cleaved fragments of PKC ϵ .

Fig. 3A and line 102-107: Authors mentioned that cleavage of PKC ϵ to the free kinase domain was blocked by caspase-7 knockdown. However, caspase-7 siRNA had only a modest effect on GFP-PKC ϵ processing. Basu et al. have shown before that PKC ϵ cleavage was inhibited by caspase-3/-7 inhibitor in MCF-7 cells, which lack caspase-3.

Comment added in text (line 167): Knock-down of Caspase 7 resulted in a 64% reduction of the cleaved PKC ϵ species, however no significant reduction in cleavage was observed with depletion of Caspase 3 (Figure 3A), suggesting a Caspase 7 dependent pathway, in agreement with observations by Basu and colleagues (Basu *et al.* JBC 2002). We would also note that the cleavage is not an equilibrium event it is unidirectional, hence it is difficult to achieve a more penetrant blockade as the residual activity ratchets towards cleavage.

Suppl. Fig. 2C, 2D & 2E: How is caspase-7 activated by nocodazole? Activation of effector caspases, such as caspase-7, requires processing by apical caspases, which are activated by apoptotic stimuli. Taxol, which stabilizes microtubules and inhibits microtubule detachments, can also induce apoptosis. Perhaps, stress caused by nocodazole treatment can cause low level of caspase-7 activation, which is sufficient for SAC regulation but not induction of apoptotic cell death. This is consistent with the slight increase in cleaved PARP by nocodazole treatment as seen in Suppl. Figure 2C.

The cell cycle dependent Caspase-7 regulation, while interesting, is beyond the scope of this current manuscript. In this study, nocodazole, taxol and STLC are tools for mitotic cell enrichment, where the cleavage of PKC ϵ is observed. Kozielski and colleagues (Kozielski *et al.* Proteomics 2008) report that apoptosis occurs after 48-72h treatment of 5 μ M STLC or 5.8 μ M Taxol through Caspase-9 activation, suggesting the shorter inhibitor treatments and lower doses used in our study are activating an apoptosis independent pathway.

Fig. 3E, right panel: The resolution of the catenated chromatids in GFP-Aurora B S227A transfected cells is low.

Image has been replaced with a higher quality image.

Although the authors have shown that Aurora B S227A mutant phenocopied the PKC ϵ caspase cleavage mutants, there was no direct demonstration that cleavage of PKC ϵ was responsible for Aurora B phosphorylation at Ser227 in vivo. The authors, however, demonstrated that phosphorylation of Topolla at S29 is diminished in cells expressing non-cleavable PKC ϵ in Suppl. Figure 3F.

Comment added in text: While direct detection of phosphorylation of Aurora B S227 by PKC ϵ during metaphase was not possible with the available reagents, these results suggest that the PKC ϵ -dependent metaphase catenation response we previously described, is manifested through phosphorylation of Aurora B S227.

We are confident that Aurora B lies downstream of PKC ϵ as in every experiment AuroraB S227A does phenocopies PKC ϵ knockdown/inhibition/non-cleavable mutants.

Fig. 5A: The authors should provide more details regarding how decatenated kDNA was quantified. Based on the DNA gel, there does not appear to have much difference in the dekatented DNA in

the presence of 0.1 and 1.0 µg WT Aurora B or 0.01 or 0.1 µg S227A mutant Aurora B. Does non-cleavable mutant PKCε affect decatenation?

These assays have been repeated many times and the images presented changed. Details of quantification have been included in the methods. While the in vitro decatenation was not assayed in the presence of the non-cleavable PKCε mutant, it should be noted that a significant increase in catenated sister chromatin pairs was observed in cells expressing this mutant, and this could be recovered with the addition of recombinant TopoIIα to the assay (Figure 1H).

Minor comments:

Pg. 10, Line 449 and 481: Reference 2 and 15 are same. **Corrected reference list**

Pg. 3, Line 146: There is an extra bracket at the end. **Removed**

Pg. 11, Line 534-537: The sentence should be corrected. **Corrected**

Pg. 25: "Supplementary Figure 5" should be "Supplementary Figure 4." **Corrected**

Reviewer #2 (Remarks to the Author):

DNA catenation needs to be resolved properly during cell division to avoid genome instability. How DNA catenation resolution is coordinated during mitotic progression is not completely understood. In the current study, Kelly et al. provide some evidence to suggest the involvement of a caspase-triggered kinase cascade in this process. Although the study contains several interesting observations, key conclusions are not sufficiently supported by actual data. Furthermore, the technical quality of many experiments in the study is low. These deficiencies cast doubt on the overall validity of their conclusions.

Major deficiencies

(1) No experiments in the current study link PKCε cleavage to Aurora B phosphorylation at S227. The authors should examine whether Aurora B pS227 signals are diminished in cells expressing non-cleavable forms of PKCε or depleted of caspase 7.

Comment added in text: While direct detection of phosphorylation of Aurora B S227 by PKCε during metaphase was not possible with the available reagents, these results suggest that the PKCε-dependent metaphase catenation response we previously described, is manifested through phosphorylation of Aurora B S227.

We are confident that Aurora B lies downstream of PKCε as in every experiment AuroraB S227A phenocopies PKCε knockdown/inhibition/non-cleavable mutants.

(2) The catenation assay used in the paper is indirect. They have used paired chromatids in Sgo1-depleted cells as a measure of decatenation failure. While reasonable, other alternative explanations exist. For example, a defect in cohesin-removal pathway would also produce more paired chromatids in Sgo1-depleted cells. This is especially important given the roles of Aurora B in cohesin removal. The authors should examine whether cohesin is indeed removed in their catenation assays. In addition, they should stain the spreads with PICH, a marker of the expected ultrafine threads connecting chromatids that are not completely decatenated.

Comment in text: The non-cleavable PKC ϵ also failed to resolve catenation in metaphase (Figure 1G, H), importantly, as judged by the rescue of non-disjoined chromatids with recombinant TopoII α , indicating that retained sister chromatid pairs was due to catenation rather than other causes of paired chromatids such as a cohesion-removal defect.

Graph axes have been relabelled 'retained sister chromatid pairs (%)'

The cohesin and PICH antibodies we tested were incompatible with methanol fixation to stain the chromosome spreads for these markers.

(3) In Figure 2E, why are there so many GFP-PKC ϵ degradation species even in the absence of TEV or rapamycin? I found it surprising that the induction of a cleavage product of PKC ϵ among the sea of similar cleavage products would produce any predictable cellular phenotypes as shown in Figure 2F.

Figure 2E has replaced with other experiments that more clearly illustrate the cleavage.

(4) The effects of Caspase 7 depletion on the cleavage of GFP-PKC ϵ are marginal. Clearly other proteases are involved. Also, can the cellular effects of Caspase 7 depletion be rescued by the expression of an RNAi-resistant transgene?

Comment added in text (line 167): Knock-down of Caspase 7 resulted in a 64% reduction of the cleaved PKC ϵ species, however no significant reduction in cleavage was observed with depletion of Caspase 3 (Figure 3A), suggesting a Caspase 7 dependent pathway, in agreement with observations by Basu and colleagues (Basu *et al.* JBC 2002).

The cellular effects of Caspase-7 knockdown could be rescued by expression of the PKC ϵ kinase domain but not expression of the full length, wild type kinase suggesting that caspase7 dependent cleavage of PKC ϵ is required for the metaphase catenation response.

(5) Results in Figure 4B are flawed. The total TopoII α blot shows uneven loading in the reactions. This needs to be corrected. It appears that the pT1460 antibody can recognize unmodified protein.

This recombinant Topo2a protein is partially phosphorylated at T1460 in the basal state; all these westerns are performed in the presence of the dephospho-antigen to eliminate any cross-recognition of the dephosphorylated protein. The western shows that this increases with the S227A mutant while the S29 site is the preferred site for the WT AuroraB which in these preparations is largely doubly phosphorylated.

(6) In Figure 5A, the in vitro decatenation assay is not convincing. Only very small amount of the catenated DNA was decatenated in the assay. The band is barely visible. In addition, TopoII α S29A and T1469A mutants need to be included in the assay as additional controls.

These assays have been repeated many times and improved images are included in this revised version. We have included also a new experiment where the mutant forms of TopoII α have been extracted from HEK293 cells and assayed for their activity (Supplementary 7F,G).

(7) The specificity of the TopoII α pS29 antibody needs to be further verified by staining cells depleted of endogenous TopoII α and expressing S29A. These cells are available as they were used in Figure 5C.

These images have been included in Figure 4F

(8) The effects of TopoII α on catenation (even with the caveat mentioned in point 2) are not compelling. Marginal differences are observed even with artificial conditions (i.e. Sgo1 depletion).

This experiment has been repeated in HEK293 cells which are induced to express GFP-TopoII α . The expression of the exogenous protein in these cells is more robust than the DLD1 cells. It is notable that the basal levels of retained sister chromatid pairs is markedly reduced in the HEK293 lines compared to DLD1, which we believed confounded the original experiment. For clarity, both experiments are included (Figure 5D and Supplementary Figure 5D).

Reviewer #3 (Remarks to the Author):

The authors report that DNA catenation in early mitosis triggers a cascade of events. In the pathway, Caspase-7 cleaves and activates PKC ϵ , which in turn acts on Aurora B to both bind topoisomerase II α and induce its phosphorylation, the purpose being to delay mitotic exit and enable DNA strands to decatenate.

Comments

Overall, the manuscript should be able to demonstrate that Caspase-7 is a key to topoisomerase II α induction of decatenation in mitosis. This could be an interesting contribution to the field, but there are many serious problems with the manuscript that prohibit publication. In general, data and images are substantially deficient, as detailed below. Further, the paper has been constructed in many places of data that are unacceptably sloppy and unconvincing. These deficiencies must be corrected.

The text does not indicate why the native PKC ϵ has to be siRNA suppressed and replaced by GFP tagged PKC ϵ in every case where this has been done. Supplemental Figure 1a shows that the native PKC ϵ is strongly detectable by western blot, so it is not clear why the native form was not subjected to western blot in Figure 1, or used for western blot without siRNA knockdown. GFP-PKC ϵ is obviously necessary for mutant analysis, but its use elsewhere remains to be explained. The authors state in Methods that GFP expression always entails native protein knockdown. It appears, however, that PKC ϵ knockdown and replacement with GFP-PKC ϵ is not always done. For example, Supplementary Figure 1K shows GFP-PKC ϵ (and mutant) expression with or without knockdown of the WT. This all needs clarification and explanation.

Supplementary figure 1K demonstrates the induction of GFP-PKC ϵ in the DLD1 cell line. A comment has been added to the text – We engineered a DLD1 cell line to express inducible, GFP-tagged PKC ϵ where the cleavage sites have been mutated (Supplementary Figure 1K). Endogenous PKC ϵ is depleted in experiments where GFP-PKC ϵ is induced to ensure results are not confounded by the dominant behaviour of the cleaved fragment. The penetrant use of this line provides consistency. However we note that for the cleavage of PKC ϵ we have demonstrated that the endogenous protein behaves exactly as the tagged. Similarly we have shown elsewhere that the effects of manipulation of tagged PKC ϵ is consistent with the effects of inhibitors on the endogenous protein indicating that what is reported is not some artefact of the tagged form.

Figure 1a. GFP-PKC ϵ +0.1% Triton X extraction yields a centrosomal signal (this is unmistakable, and a double label would confirm it). This is a major problem with respect to interpretation of the overall message of the manuscript. The text states, and logic dictates, that the signal is chromosomal. The images show otherwise. A further major problem is that the GFP tag is N-terminal, identifying the

location of the regulatory domain after caspase cleavage, while the interest is in where the catalytic domain localizes. This result is key to understanding the overall results of the manuscript, and is unacceptable.

We have previously identified a centrosomal pool of PKC ϵ (Martini et al., 2017) and comment on this in the text. We find there are sub-populations of GFP-PKC ϵ present; a centrosomal, spindle associated pool that we have previously described (Martini *et al.* Molecular Cancer Research 2018) as well as a chromatin-associated compartment that can be visualized by immunofluorescence (Figure 1A) and appears to be enriched during metaphase.

A C-terminal GFP tag of PKC ϵ is not possible due to interference with protein folding and kinase activity. However, we do observe the N-terminal regulatory domain present in the chromatin/insoluble fraction (Figure 1B) indicating that this fragment is retained in this compartment after cleavage.

Figure 1b. The TX-100 insoluble fraction is probed by western blot, but Methods neither describes the separation step, nor the TX-100 extraction procedure. Histone H3, a highly abundant protein, should not be such a light band on westerns in this figure. This indicates that there was a lack of attention in construction of the manuscript. There is no such problem in Suppl Fig 1.

Fractionation procedures have been added to the Methods section. The Histone H3 blot (Figure 1B) has been improved.

Figure 1d. Molecular mass markers should mark the full range of the proteins separated. This is not acceptable. The markers should have been the same markers used in Fig 1b.

We apologise for this oversight, the full range of molecular weight markers have been added.

Figure 2c. The large error bars suggest a variable that is not accounted for between experiments. Without the raw data one can only hope a computer derived P value gives sufficient confidence in this t-test to warrant the conclusion there is no meaningful difference. The values used for calculation should be the means of the three independent experiments. Did the authors use n=3 or n=45 for calculation? If n=3, the variance is too great to conclude much. If n=45, the calculation was done incorrectly (samples are not means). The error bars for identical sample sizes are far cleaner in Figure 3e. This indicates the authors can do the experiment correctly and this should be redone.

This experiment has been redone and the figure replaced with the new data. The values graphed are the mean of 3 independent experiments.

Figure 2e. The TEV-cleavage mutant experiment was particularly convincing in Fig 2f, but the western blot that underlies the experiment was unacceptably noisy and the TEV cleaved product was a rather weak band. Other similarly probed blots were clean. Surely the authors have a cleaner experiment to present as several experiments were run. The authors must replace this blot.

Figure 2E has been improved and replaced.

Figure 3a. Is this total cell extract or the insoluble fraction? Not stated.

Comment added to text - Knock-down of Caspase 7 resulted in a 64% reduction of the cleaved, TritonX-100 insoluble PKC ϵ species, however no significant reduction in cleavage was observed with depletion of Caspase 3 (Figure 3A). This is noted in the Figure legend.

Figure 4. Overall an impressive and convincing figure, except for the inexplicable lack of stain for the spindle in part d second row. The peptide array experiment was surprising and the kinase assay was convincing.

Images have been improved and replaced

Figure 5a. The decatenated bands are extremely difficult to see, and therefore it is difficult to agree with the quantitation presented here. An ideal decatenation assay with highly visible decatenated bands can be found, for example, in Wang L, et al. J Cell Sci 123, 806 (2010), Figure 3C. This figure should look like that example.

These assays have been repeated many times and images improved. We have included a new experiment where the mutant forms of Topolla have been extracted from HEK293 cells and included in the assay (Figure 5C).

Suppl Fig 1a. The blot could have been run and probed for full-length and fragments of the triton insoluble fraction. Why is the full length only shown for the soluble fraction?

The full length (84kDa) protein is displayed in the insoluble fraction of Supplementary figure 1A. The lack of band suggests that the majority of the endogenous PKC ϵ in this fraction is cleaved.

Suppl Fig 1f. It would be reassuring to assay the reaction rates in the kinase assays, to determine that different constructs have the same catalytic efficiency, and that the mutants are truly like the WT. As it is, one cannot conclude that catalysis is completely unaffected by mutation.

It is not possible to assert that activity against all substrates is identical. We have however shown that the activity against a conventional *in vitro* substrate is unaffected; the catalytic potential appears therefore to be unchanged.

Suppl Fig 2b. Both ICRF-193 and nocodazole are toxic to cells on long exposure. It is odd that viability over 24 hr is not influenced by exposure to these drugs. Explain.

Supplementary figure 2B (cell viability) and C (PARP cleavage) clearly demonstrates that the concentration and exposure to both ICRF193 and nocodazole are not toxic and do not induce apoptosis in the time frame employed for these experiments. This can be very different for other cell types.

Suppl Fig 2i. The absence of CREST signal in the lower two plates is inexplicable (left images). The result +ICRF-193, +nocodazole is not credible (right images). The cell chosen appears to be nearly aligned at metaphase, which is not possible at the high concentration of nocodazole used. Normally one would find a pro-metaphase array, as shown above in this figure part. Must be explained or redone. DAPI imaging is too variable throughout the figure, when it should be uniformly bright. Normally this should be a control signal that indicates microscope and camera settings were correct.

Better representative images have been included. The ICRF193+nocodazole conditions demonstrate that after acute (5min) treatment with nocodazole, these Aurora B S227A expressing

cells are able to re-localise the SAC components to the kinetochore and therefore have an intact SAC response, but in the presence of catenation, this is not triggered and the delay is bypassed. Comment added to text: Cells which express Aurora B S227A maintain a functional SAC in the presence of excess catenation as an acute (5 minute) treatment with nocodazole to disrupt the spindle results in re-localisation of MAD2/BubR1 to the kinetochore.

Suppl Fig 3a. The siNTC experiment result is simply wrong. There is a bright green signal in the color channel on the right, that indicates the presence of GFP-AurB, while the GFP-AurB channel shows no signal. Either GFP-AurB was expressed here and the color channel is correct, or GFP-AurB was not expressed, and the black and white channel is correct. The left-hand legend does not indicate GFP-AurB is present, but the label above the third lane indicates it is present, while the figure legend suggests it is absent. What is going on?

We apologise for the lack of clarity and have re-labelled these images and amended the figure legend.

In any case, evidence for overlap at the centromere is scant. An overlap signal between red and green channels should be clearly yellow, and should be evident in the first row, but is only partly evident in the third row.

Pixel intensity profiles have been added to the figure to demonstrate overlapping localisation of Aurora B and TopoII α

It is fortunate that others have found this centromeric association of Aurora B and topoII, particularly the Coelho et al paper (ref 21) cited by the authors. The Coelho paper should stand as a model to the authors of what excellent mitotic chromatin microscopic images should look like (compared to this manuscript's figure 3e for example, which again shows highly variable stain and does not give confidence that accurate quantitation can be done).

The comment is noted.

Suppl Fig3f and g. The authors use quantification of pixel intensity to graph their result. But given the high variability in microscopic image intensity presented elsewhere (remarks on Suppl Fig 2i for example), the authors must measure pixel intensity relative to another stain intensity, such as CREST or DAPI, which remains constant from cell to cell. Methods states that DAPI was used to define the area of localization for the other channel, but gave no indication it was used as a signal quantitation control.

Images have been re-analysed, normalised to DAPI.

Suppl Figs 1i, 2g. The control chromosome bridging is pretty high, suggesting that a fraction of the cells do not adequately decatenate, no matter what. This is surprising, as one function of the delay to anaphase checkpoint is to assure complete decatenation.

We consistently observe high levels of catenation in the DLD1 cell line (see also Brownlow et al, Nature Communications 2014). Investigating how these cells overcome these intertwinings is an interesting study of itself but unfortunately outside the scope of this current manuscript.

Suppl Fig 4d has very high control values for bridging, approaching the level of mutant bridging in Suppl Fig 1l. The experiment should be redone. As stated above, the function of the delay to anaphase checkpoint is to assure complete decatenation, in which case bridging (except ultrafine bridges that resolve) should be minimal. The critical diagnostic criterion should be failure of abscission if meaningful bridging persists, and cells should instead be scored for abscission (again, see Wang L, J Cell Sci 123, 806 (2010), Figure 2). Measuring abscission failure, WT cells should be negative.

We have now included data in a different cell line (HEK293), confirming the original result in DLD1 cells. Further, we have analysed the number of binucleate/aberrantly shaped nuclei in the HEK293 cells expressing GFP-TopoII α mutants and this correlates with the catenation spread assay result.

Minor points:

The authors do not mention that ICRF-193 is a topoisomerase inhibitor, nor that nocodazole is a microtubule assembly inhibitor that induces mitotic arrest, nor that RO-3306 is a cyclin dependent kinase inhibitor, information that would be useful to those not expert in the field.

This information has now been included

The Figure 2 legend title is a sentence fragment, and the first sentence in the legend is garbled (adn).

This has been corrected.

Suppl Fig 4 is mislabeled "Suppl Fig 5".

Figures are now correctly labelled

The authors vaguely state that recruitment of AurB by TopoII is "mediated by the C-terminal domain" (line 175). It is the C-terminal domain of TopoII that recruits.

Have revised text to read: ...demonstrated that Aurora B is recruited by the C-terminal domain of TopoII α , independent of the enzyme's catalytic activity

Reviewers' comments:

Reviewer #1 (Remarks to the Author):

The revised version of the manuscript is significantly improved with the replacement of poor quality figures with better figures, addition of new data and reorganization of the manuscript. The authors have adequately addressed my concerns. The manuscript should be of significant interests to the readers of Nature Communications.

Minor comments:

Suppl. Fig. 3G: It was mentioned in the Figure Legend that cells were treated with or without ICRF193 but +/- signs next to ICRF193 were not included in the Figure.

Suppl. Fig. 5C legend: pT233 should be pT232.

Suppl. Fig. 7G legend: There should be a space between (G) and Graph.

Reviewer #2 (Remarks to the Author):

Several of the major deficiencies pointed out during the initial round of review are not satisfactorily addressed. In some cases (detection of Aurora B pS227 and cohesin/PICH staining), the authors cited technical difficulties for why key experiments cannot be done. While this reviewer is sympathetic, these experiments are nonetheless not performed. Thus, my points 1 and 2 are not addressed. Addressing these crucial points would have made the story more compelling and cohesive. Without such data, the paper still suffers from obvious weaknesses. In other cases (rescue of Caspase 7 RNAi with Caspase 7 transgene and pS29 antibody specificity), incorrect experiments or results are cited, but they fail to address my points 4 and 7. I cannot support the publication of this manuscript, at least not in its present form.

We thank the reviewers for their thoughtful comments, please see below for our response.

Reviewer #1 (Remarks to the Author):

The revised version of the manuscript is significantly improved with the replacement of poor quality figures with better figures, addition of new data and reorganization of the manuscript. The authors have adequately addressed my concerns. The manuscript should be of significant interest to the readers of Nature Communications.

We are grateful for the encouragement of this reviewer.

Minor comments:

Suppl. Fig. 3G: It was mentioned in the Figure Legend that cells were treated with or without ICRF193 but +/- signs next to ICRF193 were not included in the Figure.

Corrected

Suppl. Fig. 5C legend: pT233 should be pT232.

Corrected

Suppl. Fig. 7G legend: There should be a space between (G) and Graph.

Corrected

Reviewer #2 (Remarks to the Author):

Several of the major deficiencies pointed out during the initial round of review are not satisfactorily addressed. In some cases (detection of Aurora B pS227 and cohesin/PICH staining), the authors cited technical difficulties for why key experiments cannot be done. While this reviewer is sympathetic, these experiments are nonetheless not performed. Thus, my points 1 and 2 are not addressed. Addressing these crucial points would have made the story more compelling and cohesive. Without such data, the paper still suffers from obvious weaknesses. In other cases (rescue of Caspase 7 RNAi with Caspase 7 transgene and pS29 antibody specificity), incorrect experiments or results are cited, but they fail to address my points 4 and 7. I cannot support the publication of this manuscript, at least not in its present form.

The first outstanding issue noted by Reviewer 2 is the lack of direct connection between Aurora B phosphorylation on S227 and PKC ϵ during M-phase (we had shown such a connection before with recombinant proteins *in vitro* and at cytokinesis *in vivo*). As we had indicated in our correspondence accompanying the revised version, this had been a technical challenge to link up beyond the correlative evidence of a fully congruent phenotypic relationship, reflecting the limitations on reagents for single cell analysis of this phosphorylation event. We have now found a way around this problem through fractionation of synchronised cells and have directly addressed this for mitotic cells. We demonstrate in this revised version that, upon increased catenation as a result of TopoII α inhibition, Aurora B S227 phosphorylation in a mitotic enriched population (Supplementary Figure 1A) is significantly sensitive to PKC ϵ inhibition (inhibition of PKC ϵ by either BLU577 Figure 1F or BIM1 supplementary figure 1B cause inhibition of S227 phosphorylation). We note that in asynchronous cell extracts, Aurora B phosphorylation is not sensitive to PKC ϵ inhibitors (Supplementary figure 1C), indicating that the phosphorylation of Aurora B by PKC ϵ is restricted to cells in M-phase and also likely

explains the residual inhibitor-insensitive phosphorylation in the enriched population. It remains formally possible that an element of the Aurora B S227A phenotype reflects non-mitotic events (ie not under PKC ϵ control), however the congruence of PKC ϵ inhibition/knock-down and the expression of the Aurora B S227A mutant argue for a dominant effect of the mitotic, PKC ϵ -dependent phosphorylation. This permits us to conclude that the phenotypic relationship between PKC ϵ inhibition and the expression of a non-phosphorylation competent Aurora B S227A mutant is determined by the mitotic phosphorylation of Aurora B by PKC ϵ at this site. We now include this data and discussion of its implication in the revised manuscript.

Point 2, commented on by Reviewer 2, relates to the staining of chromosomes for cohesin and PICH. We have worked extensively to stain for these antigens in the methanol fixed samples required for the chromosome spread assays and not found a condition compatible with achieving staining despite our successful, routine staining for PICH positive ultrafine bridges in anaphase cells. We would agree that staining for additional markers provides more information about the state of sister chromatids, but would argue that cohesin/PICH chromatin retention would be a correlative observation that adds limited information to the paper, for which the aim is to unpick the PKC ϵ -Aurora B relationship with respect to the ability of TOPO2A to resolve catenanes in metaphase as described. We consider it is beyond the scope of the paper to set about an antibody production programme to generate reagents that will work under these methanol fixation conditions in order to address these additional markers.

We would emphasise that the property showing dependence on PKC ϵ (i.e. the pathway under investigation) is the retention of sister chromatid linkage that is resolved *in vitro* post-extraction by recombinant TOPO2A. This is the basic assay we have described both here (Figure 3D) and previously (Brownlow *et al.*, 2014). Specifically, the increment of retained sister chromatid pairs that is reversed by treatment with recombinant TOPO2A *in vitro* provides a robust functional assay for the extent of association dependent upon catenation. This is certainly not 100% of all events and may indeed include other modes of sister chromatid association but these are not dependent on the proposed PKC ϵ -Aurora B pathway reported here.

Reviewer 2 also raises the issues around the original point 4. There are two elements to this, relating to the extent of caspase 7 inhibition of events and the rescue of caspase 7 knockdown. Regarding the former issue, we have made the case that this degree of inhibition is in line with expectations for an irreversible process of this nature - the residual caspase 7 protein post siRNA treatment will still inefficiently cleave its target(s), but they do not get stitched back together again, this is not a reversible metabolic event where incomplete enzyme inhibition can lead to an essentially complete switch in the system behaviour. Nevertheless, as the reviewer points out we can not make a definitive statement about this and hence we refer to a dominant role of caspase 7. Regarding the second issue, we have performed the most informative rescue experiment to address the action of this proposed pathway and that is to ask if the Caspase 7 substrate, ie WT PKC ϵ or the Caspase 7 product i.e. PKC ϵ kinase domain, can rescue the metaphase delay phenotype observed on knockdown of the protease. We show that in fact only the product can rescue. This adds critical information to the paper and is entirely consistent with the causative relationship proposed. We note that ectopic expression of caspase 7 (to rescue the knockdown) would address specificity of the siRNA as an siRNA effect only, the rescue with PKC ϵ kinase domain serves a more profound purpose. Notwithstanding this, as the Reviewer may be aware, ectopic expression of caspases in cells has its own intrinsic issues, induction of apoptosis being one.

Lastly in relation to point 7 we made a mistake in our response to reviewers in referring to a Figure panel we had subsequently moved to the supplementary material. The specificity of the Ser29 antiserum is very clearly illustrated in supplementary Figure 5D as submitted in the last version – we apologise for the confusion this may have caused.